# Understanding Adversarially Robust Generalization via Weight-Curvature Index

## Abstract

Despite extensive research on adversarial examples, the underlying mechanisms of adversarially robust generalization, a critical yet challenging task for deep learning, remain largely unknown. In this work, we propose a novel perspective to decipher adversarially robust generalization through the lens of the Weight-Curvature Index (WCI). The proposed WCI quantifies the vulnerability of models to adversarial perturbations using the Frobenius norm of weight matrices and the trace of Hessian matrices. We prove generalization bounds based on PAC-Bayesian theory and second-order loss function approximations to elucidate the interplay between robust generalization gap, model parameters, and loss landscape curvature. Our theory and experiments show that WCI effectively captures the robust generalization performance of adversarially trained models. By offering a nuanced understanding of adversarial robustness based on the scale of model parameters and the curvature of the loss landscape, our work provides crucial insights for designing more resilient deep learning models, enhancing their reliability and security.

## 1 Introduction

Building models to be resilient to adversarial perturbations remains an enduring challenge. An active line of research (Madry et al., 2018; Carmon et al., 2019; Andriushchenko & Flammarion, 2020; Croce et al., 2020; Gowal et al., 2020) has emphasized that adversarial training, where models are optimized to withstand worst-case perturbations, is essential to bolster robustness. While standard deep learning produces models that can generalize to unseen data well, adversarial training presents a starkly different scenario. In particular, overfitting to the training set severely harms the robust generalization of adversarial training, resulting in models that perform well on adversarial examples in the training set but poorly on those in the test set. This phenomenon, known as robust overfitting (Rice et al., 2020; Li & Li, 2023), underscores a significant gap in our understanding of deep learning generalization under adversarial settings. To mitigate robust overfitting and improve the generalization of adversarial training, various robustness-enhancing techniques have been proposed, such as data augmentation (Zhang et al., 2018; Yun et al., 2019), $\ell_2$ weight regularization (Stutz et al., 2019), early stopping (Rice et al., 2020), adversarial weight perturbation (AWP) (Wu et al., 2020), incorporating synthetically generated data (Gowal et al., 2021), sharpness-aware minimization (SAM) (Wei et al., 2023), to name a few. Nevertheless, there is still no clear understanding of why robust overfitting occurs or what factors are critical for achieving adversarially robust generalization.

A reliable indicator of adversarially robust generalization is useful for identifying the limitations of state-of-the-art methods and gaining insights to guide the development of more robust models. To better understand the generalization of deep neural networks, a variety of metrics have been proposed, including margin-based measures (Pitas et al., 2017; Jiang et al., 2018; 2019; Yang et al., 2020b), smoothness-based measures (Cisse et al., 2017), flatness-based measures (Petzka et al., 2019; Yu et al., 2021; Stutz et al., 2021; Petzka et al., 2021; Xiao et al., 2022; Kim et al., 2023; Andriushchenko et al., 2023), and gradient-norm measures (Zhao et al., 2022; Ross & Doshi-Velez, 2018; Moosavi-Dezfooli et al., 2019; Andriushchenko & Flammarion, 2020; Dong et al., 2019). Although these metrics have shown varying degrees of effectiveness in capturing the model's generalization gap for both standard and adversarial training (Neyshabur et al., 2015; Bartlett et al., 2017; Dziugaite et al., 2020; Liu et al., 2020; Keskar et al., 2016; Foret et al., 2021; Zhuang et al., 2022; Kwon et al., 2021; Wu et al., 2020), recent studies reveal that no single measure can perfectly estimate a model's robust generalization capability (Kim et al., 2024). Therefore, it is important to

develop new theoretical frameworks and more reliable indicators that can better capture a model's robust generalization capability.

**Contributions.** We introduce the *Weight-Curvature Index* (WCI), a novel metric that characterizes the robust generalization performance of adversarially trained models by leveraging the Frobenius norm of weight matrices and the trace of Hessian matrices (Definition 2.6). WCI has shown a strong correlation with the robust generalization gap and can improve adversarial robustness. In particular, the definition of the WCI is motivated by our newly derived PAC-Bayesian bound on robust generalization gap (Lemma 2.2 and Theorem 2.5), which establishes a rigorous link between model parameters, loss landscape curvature, and robust generalization performance (Section 2). Empirically, we demonstrate the strong correlation between the proposed WCI and the robust generalization performance for models during adversarial training, suggesting its potential for effective predictions of generalization gaps (Section 3). Moreover, we explore how the WCI dynamically interacts with the learning rate by introducing an algorithm that dynamically adjusts the learning rate based on the WCI during training, which improves adversarial robustness by adapting to changes in model behavior, providing insights into optimizing learning rate schedules to enhance model robustness (Section 4). We compare WCI with other popular norm-based and flatness-based measures, highlighting its superior ability to understand the complex interactions for robust generalization between weight scale and loss curvature in adversarial settings (Section 5). We conclude by summarizing our findings and discussing future directions for advancing adversarial robustness research (Section 6).

**Related Work.** Identifying a reliable metric to characterize adversarially robust generalization has been shown to be a challenging task in the existing literature. For instance, margin and smoothness measures often show strong negative correlations with the robust generalization gap (Yang et al., 2020a; Kim et al., 2024), implying that beyond a certain threshold, increasing margin and reducing smoothness may degrade robust generalization performance. In the context of norm-based measures, while studies have demonstrated that input gradient norm regularization could enhance adversarial robustness (Ross & Doshi-Velez, 2018; Huang et al., 2023), recent findings by Kim et al. (2024) suggest that a lower gradient norm does not invariably lead to improved robustness. Additionally, prior research (Jiang et al., 2019; Dziugaite et al., 2020) has shown a strong correlation between cross-entropy loss and robust generalization gap in standard training, leading to the use of early stopping based on cross-entropy thresholds to prevent overfitting. However, extending this approach to adversarial training is complicated by the varied loss functions utilized in methods like TRADES and MART. In contrast, flatness-based measures have been demonstrated to poorly correlate with robust generalization performance (Wen et al., 2024; Walter et al., 2024). Contrary to traditional assumptions, sharper minima can sometimes correlate with lower robust generalization gaps, challenging the notion that flatter minima always leads to better generalization. Walter et al. (2024) argues that flatness alone cannot fully explain adversarial robustness. In contrast, Wen et al. (2024) suggests that sharpness minimization algorithms do not only focus on reducing sharpness to achieve better generalization, calling for alternative explanations for the generalization of over-parameterized neural networks. The aforementioned literature highlights the need for alternative explanations and new theoretical frameworks to better understand robust generalization in adversarial contexts. The role of the Hessian trace in generalization has been extensively studied, such as in Ju et al. (2022), where it was shown that trace minimization correlates with improved generalization across tasks. WCI extends this concept by integrating the trace and Frobenius norm of weights, creating a robust proxy for adversarial training contexts. Regularization techniques targeting the Fisher Information Matrix trace were discussed in Jastrzebski et al. (2021), highlighting the importance of early-phase curvature control. These insights complement our findings that robust overfitting is mitigated when WCI regularization is incorporated during training. PAC-Bayesian bounds that incorporate Hessians, as explored in Golatkar et al. (2019); Patracone et al. (2024), provide a theoretical framework supporting our derivations of WCI. Specifically, the inclusion of curvature measures aligns with the PAC-Bayesian methodology, ensuring tight bounds on robust generalization errors.

## 2 WEIGHT-CURVATURE INDEX AND ITS CONNECTION TO ROBUSTNESS

This section introduces the definition of the Weight-Curvature Index (Definition 2.6) and explains its underlying connection to adversarially robust generalization (Lemma 2.2 and Theorem 2.5).

## 2.1 Bounding Robust Generalization under PAC-Bayesian Framework

Before introducing the Weight-Curvature Index, we first lay out the preliminary definition of adversarial risk, which closely connects with robust generalization and is typically used as the basis for evaluating model robustness against adversarial perturbations (Madry et al., 2018; Rice et al., 2020).

**Definition 2.1** (Adversarial Risk). Let $h_\theta : \mathcal{X} \to \mathcal{Y}$ be a classification model to be evaluated, where $\mathcal{X} \subseteq \mathbb{R}^d$ is the input space, $\mathcal{Y}$ is the output label space, and $\theta$ denotes the model parameters. Let $\mathcal{D}$ be the underlying data distribution over $\mathcal{X} \times \mathcal{Y}$, then the *adversarial risk* of $h_\theta$ is defined as:

$$\mathcal{R}_{\mathrm{adv}}(h_\theta) := \mathbb{E}_{(\boldsymbol{x}, y) \sim \mathcal{D}}\big[\ell(\theta, \boldsymbol{x} + \boldsymbol{\delta}, y)\big],$$

where $\ell(\theta, \boldsymbol{x} + \boldsymbol{\delta}, y)$ denotes the loss function such as cross-entropy loss that measures the discrepancy between the prediction of $h_\theta$ on the perturbed input $\boldsymbol{x} + \boldsymbol{\delta}$ and the ground-truth label $y$. Here, $\boldsymbol{\delta}$ denotes the worst-case perturbation with respect to $\theta$, $(\boldsymbol{x}, y)$ and the loss function $\ell$. To be more specific, let $\mathcal{B}_\epsilon(\mathbf{0}) = \{\boldsymbol{\delta}' \in \mathcal{X} : \Delta(\boldsymbol{\delta}', \mathbf{0}) \le \epsilon\}$ be the perturbation ball centered at $\mathbf{0}$ with metric $\Delta : \mathcal{X} \times \mathcal{X} \to \mathbb{R}_{\ge 0}$ and strength $\epsilon > 0$. Then, $\boldsymbol{\delta}$ is defined as the worst-case perturbation within the $\epsilon$-ball $\mathcal{B}_\epsilon(\mathbf{0})$ such that the loss function with respect to $h_\theta$ at $(\boldsymbol{x}, y)$ is maximized: $\boldsymbol{\delta} = \boldsymbol{\delta}(\theta, \boldsymbol{x}, y) = \arg\max_{\boldsymbol{\delta}' \in \mathcal{B}_\epsilon(\mathbf{0})} \ell(\theta, \boldsymbol{x} + \boldsymbol{\delta}', y)$. We follow existing literature (Madry et al., 2017; Rice et al., 2020) to consider the perturbation metric $\Delta$ as some $\ell_p$-norm bounded distance.

The following lemma, proven in Appendix A.1, establishes an adversarially robust generalization bound using the PAC-Bayesian framework, which has been pivotal in connecting the generalization of machining learning models with weight norm-based measures (McAllester, 1999; Neyshabur et al., 2017; Dziugaite & Roy, 2017; Xiao et al., 2023; Alquier et al., 2024).

**Lemma 2.2** (PAC-Bayesian Robust Generalization bound). *Let $\mathcal{D}$ be any probability distribution over $\mathcal{X} \times \mathcal{Y}$ and $\mathcal{S}$ be a set of examples drawn from $\mathcal{D}$. Consider $\mathcal{H}$ as a set of classifiers and $\mathcal{P}$ as a prior distribution over $\mathcal{H}$. Let $\lambda > 0$ and $\alpha \in (0, 1)$, then for any posterior distribution $\mathcal{Q}$ over $\mathcal{H}$ and classifier $h_\theta$, with probability at least $1 - \alpha$, the robust generalization gap is bounded by:*

$$\underbrace{\mathcal{R}_{\mathrm{adv}}(h_\theta) - \mathcal{L}_\mathcal{S}(\theta, \boldsymbol{x} + \boldsymbol{\delta}, y)}_{\textit{Robust Generalization Gap}} \le \underbrace{\mathbb{E}_{(\boldsymbol{x}, y) \sim \mathcal{D}}[\ell(\theta, \boldsymbol{x} + \boldsymbol{\delta}, y)] - \mathbb{E}_{\theta \sim \mathcal{Q}}\mathbb{E}_{(x, y) \sim \mathcal{D}}[\ell(\theta, \boldsymbol{x} + \boldsymbol{\delta}, y)]}_{\textit{Perturbation Discrepancy}}$$

$$+ \underbrace{\frac{1}{\lambda}\mathrm{KL}[\mathcal{Q}||\mathcal{P}]}_{\textit{KL Divergence}} + \underbrace{\mathbb{E}_{\theta \sim \mathcal{Q}}[\mathcal{L}_\mathcal{S}(\theta, \boldsymbol{x} + \boldsymbol{\delta}, y)] - \mathcal{L}_\mathcal{S}(\theta, \boldsymbol{x} + \boldsymbol{\delta}, y)}_{\textit{Classifier Variability}} + \underbrace{\frac{\lambda C^2}{8|\mathcal{S}|} - \frac{1}{\lambda}\ln\alpha}_{\textit{constent term}}, \quad (1)$$

*where $\mathcal{L}_\mathcal{S}(\theta, \boldsymbol{x} + \boldsymbol{\delta}, y) = \frac{1}{|\mathcal{S}|}\sum_{(\boldsymbol{x}, y) \in \mathcal{S}} \ell(\theta, \boldsymbol{x} + \boldsymbol{\delta}, y)$ is the empirical loss and $|\mathcal{S}|$ is the size of $\mathcal{S}$.*

Here, the prior distribution $\mathcal{P}$ represents our initial belief about the distribution of classifiers before observing the data, which is typically chosen based on previous knowledge or uniform assumptions across a plausible range of classifiers. The posterior distribution $\mathcal{Q}$, on the other hand, is updated based on the empirical data (including adversarial examples) observed. It represents a refined belief about the distribution of classifiers that are likely to perform well given the observed adversarial data. The process of updating from $\mathcal{P}$ to $\mathcal{Q}$ involves balancing fitting to the data against staying close to the prior beliefs to avoid overfitting, controlled by the regularization effect of the KL divergence.

Lemma 2.2 shows that the robust generalization gap can be upper bounded by three key components, denoting as *Perturbation Discrepancy*, *KL Divergence* and *Classifier Variability* respectively, plus some constant term. Equation 1 is based on the PAC-Bayesian framework, which is generic in terms of the prior distribution $\mathcal{P}$, posterior distribution $\mathcal{Q}$ and the loss function that the model aims to minimize. The first term *Perturbation Discrepancy* measures the difference between the expected loss of the classifier over the distribution of adversarial examples (perturbed inputs $\boldsymbol{x} + \boldsymbol{\delta}$) and the expected loss under the posterior distribution $\mathcal{Q}$ over classifiers. Essentially, it quantifies how much more (or less) error is induced when using adversarially perturbed examples compared to the average error across different classifiers sampled from $\mathcal{Q}$. The second term *KL Divergence* acts as a regularizer in the derived generalization bound. It measures the divergence between the posterior distribution $\mathcal{Q}$ of classifiers and the prior distribution $\mathcal{P}$. A smaller KL divergence indicates that the learned model (represented by $\mathcal{Q}$) does not stray far from our initial assumptions or beliefs about the model space (represented by $\mathcal{P}$). This term ensures that the posterior distribution does not overfit the adversarial perturbations seen in the training data. Finally, the last term *Classifier Variability* represents the variance in the performance of different classifiers sampled from the posterior distribution

$\mathcal{Q}$ on the adversarial examples. This term reflects how stable or consistent the classifiers are when they are exposed to the same perturbed inputs. A high variability might indicate that some models in the posterior distribution are significantly better or worse at handling adversarial perturbations, suggesting a potential for improving robustness by focusing on these models.

Since our objective is to establish a reliable index of robust generalization capacity, we only need to focus on terms relevant to model robustness; thus, we ignore the constant term in Equation 1. In addition, prior works have shown that reducing the *Classifier Variability* term also decreases the *Perturbation Discrepancy* term (Doshi et al., 2024; Behboodi et al., 2022; Marion, 2024; Ge et al., 2023). When the *Perturbation Discrepancy* is sufficiently small, the *Classifier Variability* can be seen as being in sync with the *Perturbation Discrepancy* term. This synchronization occurs because small perturbations typically do not cause significant deviations in the classifier's output. Essentially, if the perturbation introduced to the input data is minor, the classifier's decision boundary is less likely to be crossed, resulting in consistent and predictable outputs. Therefore, when the perturbation discrepancy is minimized, it indicates that the perturbation is within the robustness range of the classifier, thereby maintaining the stability of classification results across slightly varied inputs. Consequently, our analyses in the following sections predominantly focus on exploring the influence of the *KL Divergence* and *Classifier Variability* terms on enhancing the model's resistance to adversarial perturbations.

## 2.2 Understanding KL Divergence and Classifier Variability

So far, we have identified two key terms based on Lemma 2.2, namely *KL Divergence* and *Classifier Variability*, both of which are important for understanding robust generalization, However, since the derived bound is generic to model parameters and distributions, it remains elusive how to extract meaningful insights from the two terms to better understand the underlying mechanisms of adversarially robust generalization. Thus, we propose to incorporate *hyperprior* and adopt *second-order loss approximation* techniques to simplify them, which are explained below.

**Incoporating Hyperprior.** We introduce a hyperprior to model the standard deviation of the model parameters following Kim & Hospedales (2024). We adopt specialized hyperpriors from Sefidgaran et al. (2024) to keep prior variance invariant to parameter rescaling. We utilize a uniform hyperprior selected from a finite set of positive real numbers, ensuring precise representation with floating-point arithmetic (Wilson & Izmailov, 2020). This approach guarantees robust Bayesian inference and provides a viable framework for parameter standardization across varying scales.

The following lemma, proven in Appendix A.2, shows how *KL Divergence* can be simplified into analytical terms related to the Frobenius norm of weight matrices by incorporating the hyperprior.

**Lemma 2.3** (Otto's KL divergence (Otto et al., 2021)). *Assume the prior distribution $\mathcal{P}$ is Gaussian with zero mean and covariance $(\sigma_{\mathcal{P}}^2 \mathbf{I})$, the posterior distribution $\mathcal{Q}$ is Gaussian with mean $\theta$ and covariance $(\sigma_{\mathcal{Q}}^2 \mathbf{I})$, and the prior variance is equal to the posterior variance layerwise, where $\sigma_k^2$ denotes both variances for the $k$-th layer. Then, the KL Divergence term can be simplified as:*

$$\frac{1}{\lambda} \mathrm{KL}[\mathcal{Q}||\mathcal{P}] = \sum_k \frac{\|\mathbf{W}_k\|_F^2}{2\lambda \sigma_k^2} + \text{const.} \tag{2}$$

*Here, $\mathbf{W}_k$ is the weight matrix of the $k$-th layer and $\|\mathbf{W}_k\|_{\mathrm{F}}$ denotes its Frobenius norm.*

Note that in Lemma 2.3, when we fix the prior variances, the KL divergence term is proportional to the squared Frobenius norm of parameters. However, since we introduced the special prior, we can arbitrarily change the prior variance after training, thereby controlling the KL divergence. To minimize the KL divergence, the variances of the prior and posterior distributions with respect to the weights for each network layer are assumed to be equal. These variances reflect the spread of the weight values and are key to understanding model robustness; larger variances in the posterior suggest a model that is more sensitive to input perturbations. This alignment reflects the weight value spread, crucial for assessing model robustness, where larger posterior variances suggest greater sensitivity to input perturbations—a key consideration in adversarial settings. Employing Gaussian priors and posteriors, as supported by the PAC-Bayesian framework (Mbacke et al., 2023; Jin et al., 2022), maintains the soundness of our theoretical results.

**Second-Order Loss Approximation.** To integrate PAC-Bayesian theory with the Hessian matrix of the loss landscape, we employ a second-order loss approximation, building on insights from recent research (Li & Giannakis, 2024; Xie et al., 2024; Wen et al., 2024). The following lemma, proven in Appendix A.3, connects the *Classifier Variability* term to the trace of the Hessian matrices.

For any $\theta'$ sampled from the posterior distribution $\mathcal{Q}$, we assume that the empirical robust loss at $\theta'$ can be approximated by: $\mathcal{L}_S(\theta', \boldsymbol{x} + \boldsymbol{\delta}(\theta'), y) \approx \mathcal{L}_S(\theta, \boldsymbol{x} + \boldsymbol{\delta}(\theta), y) + \frac{1}{2}\Delta\theta^\top\nabla_\theta^2\mathcal{L}_S(\theta, \boldsymbol{x} + \boldsymbol{\delta}(\theta), y)\Delta\theta$, where we explicit write out the dependence on the model parameters in the $\boldsymbol{\delta}$ notation to avoid confusion. The first-order term can be discarded because $\theta$ is considered to be at or near a stationary point of the robust loss function, which is a common setting considered in prior literature on deep learning generalization (Stephan et al., 2017; Keskar et al., 2016).

**Lemma 2.4** (Hessian-based Variability (Foret et al., 2021)). *Assume that the model parameters $\theta$ are converging toward a stationary distribution and the empirical loss $\mathcal{L}_S(\theta, \boldsymbol{x} + \boldsymbol{\delta}, y)$ can be approximated using second-order Taylor expansion. Then, the relationship between the expected variability in the classifier's performance and the curvature of the loss landscape is given by:*

$$\mathbb{E}_{\theta \sim \mathcal{Q}}[\mathcal{L}_S(\theta, \boldsymbol{x} + \boldsymbol{\delta}, y)] - \mathcal{L}_S(\theta, \boldsymbol{x} + \boldsymbol{\delta}, y) \approx \frac{1}{2}\sum_k \text{Tr}(\mathbf{H}_k) \cdot \sigma_k^2, \quad (3)$$

*where $\text{Tr}(\mathbf{H}_k)$ denotes the accumulation of the diagonal elements of the Hessian matrix of the empirical loss with respect to the weight matrix $\mathbf{W}_k$, and $\sigma_k^2$ is the variance associated to $\mathbf{W}_k$.*

In Lemma 2.4, we assume that during adversarial training, model parameters $\theta$ converge towards regions where the first-order derivatives of the loss function are negligible, justifying the use of a second-order Taylor expansion for the adversarial loss landscape. This assumption is well-supported by studies such as those by Dinh et al. (2017) and Yao et al. (2018), which suggest that deep learning models frequently settle in flatter regions of the loss surface where the gradients are small, thus allowing a quadratic approximation to provide a reliable representation of local variations in loss. Such conditions are crucial in adversarial training, where understanding and stabilizing the model against small perturbations directly influences its robustness. The second-order approximation, therefore, not only simplifies the mathematical analysis but also aligns closely with the empirical behavior of models under adversarial conditions, making it a practical and theoretically sound approach.

## 2.3 INTRODUCING WEIGHT-CURVATURE INDEX

Putting pieces together, the following theorem, proven in Appendix A.4, establishes an upper bound on the *KL Divergence* and *Classifier Vulnerability* terms in Equation 1, which is related to the Frobenius norm of layer-wise model weights and the trace of the corresponding Hessian matrices.

**Theorem 2.5.** *Under the same settings as in Lemmas 2.2-2.4, we have (up to some constant terms):*

$$\frac{1}{\lambda}\text{KL}[\mathcal{Q}||\mathcal{P}] + \mathbb{E}_{\theta \sim \mathcal{Q}}[\mathcal{L}_S(\theta, \boldsymbol{x} + \boldsymbol{\delta}, y)] - \mathcal{L}_S(\theta, \boldsymbol{x} + \boldsymbol{\delta}, y) \approx \frac{1}{\sqrt{\lambda}}\sum_k \sqrt{\|\mathbf{W}_k\|_\text{F}^2 \cdot \text{Tr}(\mathbf{H}_k)}. \quad (4)$$

Note that the PAC-Bayesian robust generalization bound is generic, meaning that Equation 1 holds for any prior and posterior distributions ($\mathcal{P}$ and $\mathcal{Q}$). Therefore, we choose the layerwise variances $\sigma_k$ to minimize the bounds on the sum of *KL Divergence* and *Classifier Vulnerability* in the proof of Theorem 2.5. According to Equation 4, irrespective of the value of $\lambda$ that achieves the infimum in the PAC-Bayesian bound, a smaller value of the combined metric $\sum_k \sqrt{\|\mathbf{W}_k\|_\text{F}^2 \cdot \text{Tr}(\mathbf{H}_k)}$ implies a tighter robust generalization bound. Below, we lay out the formal definition of the proposed WCI.

**Definition 2.6** (Weight-Curvature Index). The *Weight-Curvature Index* is defined as:

$$\text{WCI} := \sum_k \sqrt{\|\mathbf{W}_k\|_\text{F}^2 \cdot \text{Tr}(\mathbf{H}_k)}, \quad (5)$$

where $\mathbf{W}_k$ is the weight matrix of the $k$-th layer and $\|\mathbf{W}_k\|_F$ denotes its Frobenius norm, while $\mathbf{H}_k$ is the Hessian matrix of the loss function with respect to $\mathbf{W}_k$ and $\text{Tr}(\mathbf{H}_k)$ stands for its trace.

*Remark* 2.7. According to Equation 5, WCI is scaled by the magnitude of weight matrices, ensuring the metric invariant to parameter rescaling (Mueller et al., 2024). According to Lemma 2.2 and Theorem 2.5, a larger value of WCI indicates a higher robust generalization gap, suggesting more

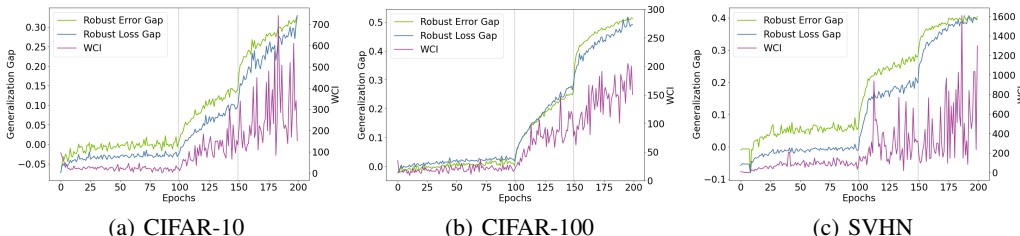

(a) CIFAR-10        (b) CIFAR-100        (c) SVHN

Figure 1: Learning curves of the Weight-Curvature Index of standard adversarial training with respect to robust generalization gaps (a) on CIFAR-10, (b) on CIFAR-100, and (c) on SVHN.

vulnerability to adversarial perturbations. The index characterizes the interaction between the scale of the model's parameters and the curvature of the loss landscape, capturing both norm-based and flatness-based measures, which offers a more comprehensive framework for understanding robust generalization. On the one hand, WCI incorporates the Frobenius norm of the weight matrix, but it extends beyond traditional norm-based approaches by integrating the trace of the Hessian matrix. Such a composite index addresses the limitations of pure norm-based metrics—such as lack of interpretability—by offering a clear, intuitive relationship where a lower WCI value signifies better performance. On the other hand, WCI is also linked to flatness-based measures, as it leverages the trace of the Hessian to capture the flatness of the loss landscape, which is critical for understanding robust generalization in adversarial contexts. The dual nature of WCI, combining both norm and flatness perspectives, allows it to better explain the robust generalization gap, positioning it as an effective tool for enhancing the robustness of neural networks in adversarial settings.

## 3 VALIDATING THE CONNECTION OF WCI AND ROBUST GENERALIZATION

In Section 2, we introduced the Weight-Curvature Index and explained how it is derived and connects with robust generalization from a theoretical perspective. Nevertheless, the proofs of the theoretical connection rely on assumptions, such as Gaussian hyperprior and second-order loss approximation, that may not always hold for models in practice. Therefore, this section further studies the relationship between WCI and robust generalization gap for adversarially trained models by conducting a series of experiments inspired by the methodology and findings of Rice et al. (2020).

In particular, we first train a ResNet-18 model on CIFAR-10, CIFAR-100, and SVHN using standard adversarial training and subsequently compute the WCI alongside the robustness and robust generalization gap. See Appendix B for detailed experimental settings. The results are shown in Figure 1. Figure 1(a) compares the generalization gap—measured through robust loss and error gaps—with WCI across 200 training epochs on CIFAR-10. We can observe that the WCI and generalization gap exhibit a consistent trend, particularly during periods of robust overfitting, and the strong positive correlation between WCI and generation gap is numerically proved using different seeds (see Appendix C.1 for detailed results and discussions). The period of robust overfitting is characterized by substantial modifications in the loss surface, reflected by fluctuations in the Hessian matrix, affecting the WCI. Despite these perturbations, a persistent alignment between WCI and the trends in robust error and loss is observed, highlighting the efficacy of WCI as a metric for monitoring robust generalization. We can see exactly the same phenomenon for CIFAR-100 shown in Figure 1(b). In addition, we perform the same experiment on SVHN. Results are depicted in Figure 1(c). The real-world complexity of SVHN might introduce more variability in the loss surface rather than CIFAR-10, leading to less smooth optimization and greater fluctuations in WCI. However, this does not affect the consistent trend of WCI curves. The strong correlation observed between WCI and the robust generalization gap reinforces the utility of WCI as a critical indicator of model performance under adversarial conditions, particularly in the context of robust training scenarios. We further examine the relationship between WCI and the robust generalization gap across various regularization and data augmentation techniques. See Appendix C.2 for detailed results.

Our empirical findings affirm the theoretical underpinnings of WCI as a reliable indicator of a model's robustness and generalization capacity. Higher WCI values are consistently associated with larger robustness losses and error gaps, indicating diminished generalization performance. These re-

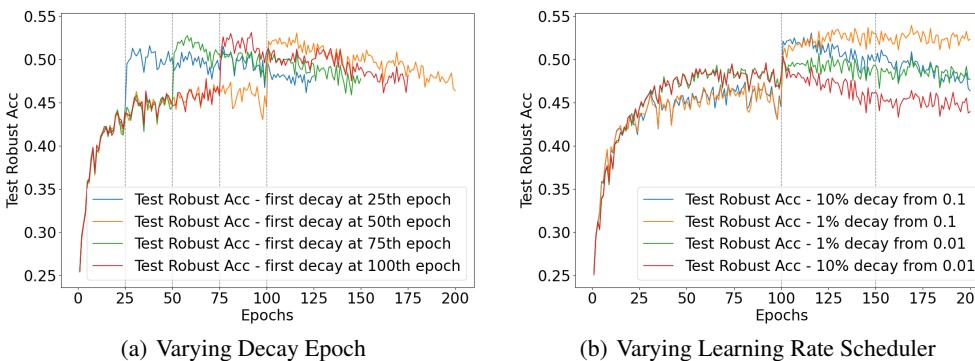

(a) Varying Decay Epoch      (b) Varying Learning Rate Scheduler

Figure 2: (a) Learning curves with the same learning rate scheduling strategy but decay at varying epochs. (b) Learning curves with different learning rate scheduling strategies, where models are trained with initial rates $\{0.1, 0.01\}$ and decay rates $\{10\%, 1\%\}$.

sults suggest that monitoring WCI during training can provide valuable insights into the robustness and generalization potential of neural networks, particularly in adversarial training settings.

## 4   IMPLICATIONS OF WCI ON MITIGATING ROBUST OVERFITTING

To ensure the model's robust generalization performance, it is essential to maintain the value of WCI to be sufficiently small. It is obvious from Equation 5 that the Frobenius norm can be easily controlled to be small, but the size of the Hessian matrix of the loss function is not easy to control. Liu et al. (2021) showed that one can tune the learning rate such that the KL divergence between the learned distribution by SGD and the posterior is minimized. Therefore, we explore the relationship between the Weight-Curvature Index and the learning rate of adversarial training algorithms in this section. More specifically, we first study the learning rate decay strategy and then introduce a new algorithm that dynamically adjusts the learning rate based on the Weight-Curvature Index.

**Learning Rate Adjustment using WCI.** Selecting proper learning rates is crucial for training deep neural networks as it sets the parameter update step size, affecting convergence speed and model generalization (Smith & Topin, 2019). Liu et al. (2021) found that while tuning the learning rate can minimize KL divergence, this only holds for smaller rates; larger rates do not require adjustments. For a specific learning rate decay strategy, we need to consider when to decay, the initial learning rate, and its decay rate. Therefore, we conducted a series of experiments under the same conditions described in Rice et al. (2020), varying the learning rates to observe their effects during the training process. Figure 2(a) shows learning curves with the same decay strategies but different decay timings at the 25th, 50th, 75th, and 100th epochs. Each decay reduces the learning rate to $10\%$ of its original value for 50 epochs, followed by similar decays. These curves follow similar trends, indicating that the timing of learning rate decay does not significantly affect robust overfitting, suggesting no need for adjustments to the decay period from the initial experiment. Figure 2(b) explores different initial learning rates and decay rates, applying a uniform decay at the 100th epoch. We tested initial rates of $\{0.1, 0.01\}$ and decay rates of $\{10\%, 1\%\}$. The best performance came from an initial rate of $0.1$ and a decay to $0.01$, although its benefit decreased over time due to robust overfitting. Our learning rate decay strategy maintains an initial rate of $0.1$ for the first 100 epochs, then decays to $0.01$, adjusting thereafter based on the value of the Weight-Curvature Index.

Our work suggests that when the learning rate decays to a smaller value during the later stages of training, a dynamic adjustment based on the WCI should be employed instead of relying on a static learning rate. Stephan et al. (2017) provided rigorous mathematical evidence demonstrating that the learning rate, when dynamically adjusted, is inversely proportional to the trace of the Hessian matrix. Building upon this, Kim et al. (2024) highlighted that efforts to maximize the margin and minimize smoothness adversely impact robust generalization performance beyond a certain threshold. Therefore, we propose to preset a threshold: when WCI exceeds this threshold, we dynamically adjust the learning rate based on WCI. In particular, we design a straightforward algorithm to em-

---

**Algorithm 1** Dynamic Learning Rate Adjustment Based on WCI

1: Initialize learning rate $\eta = 0.1$ ▷ Initial learning rate of adversarial training
2: **for** $epoch = 1$ to $100$ **do**
3:     Update model parameters ▷ Standard parameter update in initial epochs
4: **end for**
5: Set learning rate $\eta = 0.01$ ▷ Learning rate decay after initial epochs
6: **while** training continues **do**
7:     Update model parameters
8:     Compute WCI based on Equation 5 ▷ Use WCI as adaptation criterion
9:     **if** WCI exceeds threshold **then**
10:         $\eta \leftarrow \frac{\eta}{\text{WCI}}$ ▷ Adjust learning rate dynamically based on WCI
11:     **end if**
12: **end while**

---

Figure 3: Learning curves of WCI in standard adversarial training on CIFAR-10 with dynamic learning rate adjustment with (a) different thresholds and (b) with a trendline post-100th epoch.

pirically validate the dynamic interaction between WCI and the learning rate. The pseudocode of our learning rate adjustment strategy is detailed in Algorithm 1.

**Experiments.** Built on Algorithm 1, we further conduct experiments to validate the effectiveness of dynamic learning rate adjustment based on WCI in mitigating robust overfitting. Our experiments adhere to the training configuration prescribed by Rice et al. (2020). We conducted experiments using a range of WCI thresholds from 10 to 100 (in increments of 10) to explore the impact of these values on robust generalization. As illustrated in Figure 3, the results demonstrate a consistent upward trend in robust test accuracy when employing dynamic learning rate adjustments, in contrast to the downward trend observed with a static learning rate. This emphasizes the importance of dynamically adjusting the learning rate in response to the WCI to improve model generalization. Figure 3(a) highlights the advantages of using our dynamic learning rate adjustment strategy. We observe that the mitigation of robust overfitting is not sensitive to the value of the selected threshold. Thus, we fix the threshold as 100 for simplicity in the following discussions (see Figure 8 in Appendix C.3 for similar results with other thresholds). Furthermore, Figure 3(b) focuses on experiments with a threshold of 100, where a linear fit of the results post-100 epochs reveals that robust accuracy remains stable, without any decline. This supports our hypothesis that robust generalization results from a combined effect of the weight matrix norm, the trace of the Hessian matrix, and the learning rate. The consistency of these findings across different thresholds further validates our understanding of the underlying mechanisms governing robust generalization.

Table 1 compares our method with other learning rate scheduling strategies for adversarial training testing by PGD attack Madry et al. (2018). The robustness of both *Final* and *Best* models produced using our WCI-based strategy is the best, with the smallest robust generalization gap and comparable standard performance. We also compare the learning rate curves and WCI curves before and after tuning (See Figure 9 in Appendix C.4). Our comparison results suggest that incorporating WCI into the training process allows for a more nuanced control over model updates, which is particularly beneficial for adversarial training scenarios where robustness is as critical as accuracy. Also,

Table 1: Comparison of adversarial training with various learning rate scheduling strategies. Here, *Best* refers to the model with the highest test robust accuracy during training, while *Final* refers to the model at the last training epoch. For each setting, we report both robust and standard accuracies.

| Learning Rate Scheduler | Rob Acc (%) | | | Std Acc (%) | |
|---|---|---|---|---|---|
| | Final | Best | Diff | Final | Best |
| Baseline Piecewise (Rice et al., 2020) | 46.35 | 53.04 | 6.69 | 84.81 | 81.91 |
| Cosine (Carmon et al., 2019) | 45.01 | 51.04 | 6.03 | 84.51 | 82.19 |
| Cyclic (Wong et al., 2020) | 50.85 | 52.02 | 1.17 | **84.87** | **83.91** |
| Piecewisezoom (Rice et al., 2020) | 47.62 | 51.92 | 4.30 | 84.02 | 83.32 |
| Piecewisezoom-long (Rice et al., 2020) | 48.67 | 49.66 | 0.99 | 79.21 | 78.36 |
| Smartdrop (Rice et al., 2020) | 40.37 | 49.18 | 8.81 | 81.60 | 78.23 |
| Our WCI-based (test with PGD) | **52.98** | **53.80** | **0.82** | 83.42 | 83.65 |
| Our WCI-based (test with AutoAttack) | 47.68 | 48.75 | 1.07 | 82.65 | 82.05 |

we employ AutoAttack (Croce & Hein, 2020) for a more rigorous evaluation of model robustness. While accuracy decreased by approximately $5\%$ on CIFAR-10, the overall robustness trends and generalization indicators remained consistent, demonstrating the reliability of our approach. We believe these results highlight the scalability of our method while ensuring robustness across a variety of adversarial attack strategies. Our findings demonstrate that incorporating WCI into adversarial training effectively reduces overfitting and boosts model robustness without sacrificing standard accuracy. This innovative approach strikes a crucial balance between accuracy and security in neural network training, significantly contributing to adversarial machine learning research.

## 5 FURTHER DISCUSSIONS

**Role of WCI during Training.** Dziugaite & Roy (2017) showed that various stages of model training are affected by different indicators. Therefore, to better illustrate the role of WCI, we decompose the effects of the weight matrix norms and the trace of Hessian matrices in WCI. Specifically, we employ the Cauchy-Schwarz inequality to establish the following upper bound, proven in Appendix A.5, such that we can quantitatively study the role of different factors in WCI:

$$\text{WCI} = \sum_k \sqrt{\|\mathbf{W}_k\|_{\text{F}}^2 \cdot \text{Tr}(\mathbf{H}_k)} \leq \sqrt{\left(\sum_k \|\mathbf{W}_k\|_{\text{F}}^2\right)} \cdot \sqrt{\left(\sum_k \text{Tr}(\mathbf{H}_k)\right)}. \tag{6}$$

Equation 6 enables us to separate the analyses of the impact of weight norms and the trace of Hessian matrics on robust generalization. Figure 4(a) illustrates the roles of the Frobenius norm and the trace of the Hessian matrix during different training stages, whereas Figure 4(b) shows the overall changes of WCI. During the initial training stages, the Frobenius norm of the weight matrix is the most critical factor, as it determines the scale of the model's parameters. After the learning rate decay in the 100th epoch, the trace of the Hessian matrix becomes the most important factor, as it influences the model's overfitting. Our Weight-Curvature Index captures the interaction between these two factors, providing a comprehensive understanding of the model's generalization ability.

**Connection of WCI with Robustness-enhancing Techniques.** Understanding how existing robustness-enhancing techniques interact with the Weight-Curvature-Index can provide deeper insights into their efficacy in improving model robustness. For instance, $\ell_2$ weight regularization (Stutz et al., 2019) reduces the Frobenius norm of weights ($\|\mathbf{W}_k\|_{\text{F}}$), which can lower WCI and smooth the loss landscape, enhancing the model's stability against adversarial inputs and boosting robustness. Data augmentation techniques (Zhang et al., 2018; Yun et al., 2019) enhance the diversity and complexity of training data, indirectly affecting the model's behavior and facilitating exploration of flatter loss landscape regions, which might reflect in improved WCI-based robustness assessments by reducing $\text{Tr}(\mathbf{H}_k)$. Similarly, incorporating synthetically generated data (Gowal et al., 2021) broadens the model's exposure to diverse training examples, helping achieve an optimal

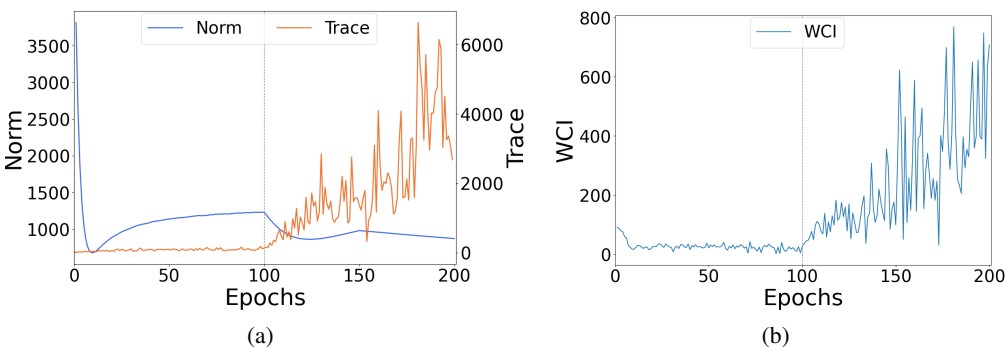

(a)    (b)

Figure 4: Illustration of the roles of the Frobenius norm and the trace of the Hessian matrix during different training stages (a), and the overall changes of the Weight-Curvature Index (b).

balance of weight magnitudes and curvature for enhanced robustness, as gauged by WCI. Adversarial weight perturbation (Wu et al., 2020) strategically modifies $\|\mathbf{W}_k\|_F$ and $\text{Tr}(\mathbf{H}_k)$, adjusting WCI values to direct the model towards parameter space regions with potentially flatter curvature and smaller weight norms, thereby improving adversarial robustness. Sharpness-aware minimization (Wei et al., 2023) targets flatter regions in the loss landscape by minimizing the maximal sharpness around current parameters, reducing the WCI by reducing $\text{Tr}(\mathbf{H}_k)$, and promoting better generalization in adversarial settings. However, Stochastic Weight Averaging (SWA) and Sharpness-Aware Minimization (SAM) focus on flattening the loss landscape. Flatness-based measures tend to exhibit poor correlations with the robust generalization gap (Kim et al., 2024). The WCI provides a specific metric that combines weight magnitude and curvature, which may offer different insights, and we have confirmed that WCI is strongly correlated with gap generation. Wen et al. (2024) suggested that sharpness minimization algorithms do not only minimize sharpness to achieve better generalization, which calls for the search for other explanations for the generalization of over-parameterized neural networks. Through these analyses, WCI's role as a crucial metric becomes apparent, especially in gauging the underlying mechanisms of various robustness-enhancing techniques aimed at improving the adversarial durability of machine learning models.

## 6    CONCLUSION AND FUTURE WORK

We introduced WCI, a novel metric that strongly connects with the robust generalization capabilities of adversarially trained models. Our work opens avenues for future research, particularly in applying WCI to mitigate robust overfitting and exploring its potential in further refining adversarial defenses. Although we demonstrate the effectiveness of WCI-based learning rate adjustments in mitigating robust overfitting, a limitation of integrating WCI in adversarial training is its high computation demand for computing the trace of the Hessian matrices (Appendix D). Designing effective optimization techniques to lower the costs of WCI computations and studying how to leverage WCI in other algorithms for building robust models are interesting future directions. In addition, large fluctuations in WCI learning curves exist, which remain poorly understood and add an additional layer of unpredictability to utilizing WCI for training adjustments. Future research can investigate the root causes of such fluctuations, study how to stabilize the WCI measures during training, and develop more reliable learning rate adjustment strategies for further robustness improvement.

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

# A  DETAILED PROOFS OF MAIN THEORETICAL RESULTS

## A.1  PROOF OF LEMMA 2.2

To prove Lemma 2.2, we need to make use of the following lemma. Lemma A.1 characterizes a fundamental PAC-Bayes bound, known as Catoni's bound (Alquier et al., 2024), but is adapted for adversarially robust learning, the center question we focused on in this work.

**Lemma A.1.** *Let $\lambda > 0$, $\alpha \in (0, 1)$, and let $\mathcal{D}$ be any distribution. Let $\mathcal{H}$ be a set of classifiers and $\mathcal{P}$ be a prior distribution supported by $\mathcal{H}$. For any $h_\theta$ in $\mathcal{H}$, $\theta$ denotes the set of parameters that determine the behavior of $h_\theta$. For a training set $\mathcal{S}$ of $|\mathcal{S}|$ samples $(\boldsymbol{x}, y)$ drawn from $\mathcal{D}$, where $\boldsymbol{x}$ is the input data and $y$ is the corresponding label. For any posterior distribution $\mathcal{Q}$ over $\mathcal{H}$, we define:*

- *The expected loss under distribution $\mathcal{D}$ for parameters:*

$$\mathbb{E}_{\theta \sim \mathcal{Q}} \mathbb{E}_{(\boldsymbol{x}, y) \sim \mathcal{D}} \big[ \ell(\theta, \boldsymbol{x} + \boldsymbol{\delta}, y) \big],$$

- *The empirical estimate of the loss on the training set $\mathcal{S}$:*

$$\mathbb{E}_{\theta \sim \mathcal{Q}} \big[ \mathcal{L}_\mathcal{S}(\theta, \boldsymbol{x} + \boldsymbol{\delta}, y) \big].$$

*The following bound is satisfied with probability at least $1 - \alpha$:*

$$\mathbb{E}_{\theta \sim \mathcal{Q}} \mathbb{E}_{(\boldsymbol{x}, y) \sim \mathcal{D}} \big[ \ell(\theta, \boldsymbol{x} + \boldsymbol{\delta}, y) \big] \leq \mathbb{E}_{\theta \sim \mathcal{Q}} \big[ \mathcal{L}_\mathcal{S}(\theta, \boldsymbol{x} + \boldsymbol{\delta}, y) \big] + \frac{\lambda C^2}{8|\mathcal{S}|} + \frac{\mathrm{KL}[\mathcal{Q}\|\mathcal{P}] + \ln \frac{1}{\alpha}}{\lambda}.$$

*Proof.* For the sake of completeness, we present the proof of Lemma A.1 first, which is based on the PAC-Bayesian theorem (Alquier et al., 2024). The bound is derived by applying the PAC-Bayesian theorem to the expected loss under distribution $\mathcal{D}$ and the empirical estimate of the loss on the training set $\mathcal{S}$. Catoni's bound shows that for any $\lambda > 0$, any $\alpha \in (0, 1)$,

$$\mathbb{P}_\mathcal{S} \left( \forall \mathcal{Q} \in \mathcal{P}(\Theta), \mathbb{E}_{\theta \sim \mathcal{Q}}[R(\theta)] \leq \mathbb{E}_{\theta \sim \mathcal{Q}}[r(\theta)] + \frac{\lambda C^2}{8|\mathcal{S}|} + \frac{\mathrm{KL}(\mathcal{Q}\|\mathcal{P}) + \ln \frac{1}{\alpha}}{\lambda} \right) \geq 1 - \alpha.$$

Define the empirical loss $\mathbb{E}_{\theta \sim \mathcal{Q}} \big[ \ell(\theta, \boldsymbol{x} + \boldsymbol{\delta}, y) \big]$ as the average loss over the training set with perturbations $\boldsymbol{\delta}$, and the expected loss $\mathbb{E}_{\theta \sim \mathcal{Q}} \mathbb{E}_{(\boldsymbol{x}, y) \sim \mathcal{D}} \big[ \ell(\theta, \boldsymbol{x} + \boldsymbol{\delta}, y) \big]$ as the average loss across the distribution $\mathcal{D}$ considering the same perturbations. Applying Catoni's bound involves a theoretical result that relates the true risk $R(\theta) = \mathbb{E}_{(\boldsymbol{x}, y) \sim \mathcal{D}} \big[ \ell(\theta, \boldsymbol{x} + \boldsymbol{\delta}, y) \big]$ of a hypothesis $\theta$ and its empirical risk $r(\theta) = \mathcal{L}_\mathcal{S}(\theta, \boldsymbol{x} + \boldsymbol{\delta}, y)$ on a finite sample set.

The bound states that with probability at least $1 - \alpha$, the following inequality holds for all probability distributions $\mathcal{Q}$ on the hypothesis space $\Theta$ induced by $\mathcal{P}$:

$$\mathbb{E}_{\theta \sim \mathcal{Q}}[R(\theta)] \leq \mathbb{E}_{\theta \sim \mathcal{Q}}[r(\theta)] + \frac{\lambda C^2}{8|\mathcal{S}|} + \frac{\mathrm{KL}(\mathcal{Q}\|\mathcal{P}) + \ln \frac{1}{\alpha}}{\lambda},$$

where $C$ is a bound on the loss function $\mathcal{L}$, and $\mathrm{KL}(\mathcal{Q}\|\mathcal{P})$ represents the Kullback-Leibler divergence from the posterior $\mathcal{Q}$ to the prior $\mathcal{P}$.

For the adversarial setting, this bound becomes particularly useful in assessing how well a model trained with adversarial examples (represented by $\boldsymbol{\delta}$) can generalize from its empirical loss on training data to its expected performance on the overall distribution. The bound provides a trade-off between the empirical loss and the expected loss, with the KL divergence and classifier variability terms contributing to the generalization error. Since $\mathcal{Q}$ is a distribution over classifiers, integrating the Catoni's bound over $\mathcal{Q}$ yields:

$$\mathbb{E}_{\theta \sim \mathcal{Q}} \mathbb{E}_{(\boldsymbol{x}, y) \sim \mathcal{D}} \big[ \ell(\theta, \boldsymbol{x} + \boldsymbol{\delta}, y) \big] \leq \mathbb{E}_{\theta \sim \mathcal{Q}} \big[ \mathcal{L}_\mathcal{S}(\theta, \boldsymbol{x} + \boldsymbol{\delta}, y) \big] + \frac{\lambda C^2}{8|\mathcal{S}|} + \frac{\mathrm{KL}(\mathcal{Q}\|\mathcal{P}) + \ln \frac{1}{\alpha}}{\lambda}.$$

Therefore, we complete the proof of Lemma A.1. $\square$

*Proof of Lemma 2.2.* Using Definition 2.1 and Lemma A.1, we can immediately derive the adversarial risk bound:

$$
\begin{aligned}
\mathcal{R}_{\text{adv}}(h_\theta) :=& \mathbb{E}_{(\boldsymbol{x},y)\sim\mathcal{D}}\big[\ell(\theta,\boldsymbol{x}+\boldsymbol{\delta},y)\big] \\
=& \mathbb{E}_{(\boldsymbol{x},y)\sim\mathcal{D}}\big[\ell(\theta,\boldsymbol{x}+\boldsymbol{\delta},y)\big] - \mathbb{E}_{\theta\sim\mathcal{Q}}\mathbb{E}_{(\boldsymbol{x},y)\sim\mathcal{D}}\big[\ell(\theta,\boldsymbol{x}+\boldsymbol{\delta},y)\big] \\
& + \mathbb{E}_{\theta\sim\mathcal{Q}}\mathbb{E}_{(\boldsymbol{x},y)\sim\mathcal{D}}\big[\ell(\theta,\boldsymbol{x}+\boldsymbol{\delta},y)\big] \\
\leq& \mathbb{E}_{(\boldsymbol{x},y)\sim\mathcal{D}}\big[\ell(\theta,\boldsymbol{x}+\boldsymbol{\delta},y)\big] - \mathbb{E}_{\theta\sim\mathcal{Q}}\mathbb{E}_{(\boldsymbol{x},y)\sim\mathcal{D}}\big[\ell(\theta,\boldsymbol{x}+\boldsymbol{\delta},y)\big] \\
& + \mathbb{E}_{\theta\sim\mathcal{Q}}\big[\mathcal{L}_{\mathcal{S}}(\theta,\boldsymbol{x}+\boldsymbol{\delta},y)\big] - \mathcal{L}_{\mathcal{S}}(\theta,\boldsymbol{x}+\boldsymbol{\delta},y) \\
& + \lambda^{-1}\text{KL}[\mathcal{Q}||\mathcal{P}] + \frac{\lambda C^2}{8|\mathcal{S}|} + \mathcal{L}_{\mathcal{S}}(\theta,\boldsymbol{x}+\boldsymbol{\delta},y) - \lambda^{-1}\ln\alpha,
\end{aligned}
$$

which completes the proof of Lemma 2.2. □

## A.2 PROOF OF LEMMA 2.3

*Proof.* In the context of adversarial machine learning, the Kullback–Leibler (KL) divergence measures how a model's distribution $\mathcal{Q}$, representing the learned classifiers, diverges from a prior distribution $\mathcal{P}$ under adversarial conditions. Specifically, this divergence can be adapted to account for the added complexity introduced by adversarial perturbations to the input data.

The KL divergence term is adapted for adversarial conditions as follows:

$$
\text{KL}[\mathcal{Q}||\mathcal{P}] = \sum_k \left[ \ln\left(\frac{\sigma_{k_\mathcal{P}}}{\sigma_{k_\mathcal{Q}}}\right) + \frac{\|\mathbf{W}_k\|_{\text{F}}^2 + \sigma_{k_\mathcal{Q}}^2}{2\sigma_{k_\mathcal{P}}^2} \right] + \text{const.}
$$

Here, $\mathbf{W}_k$ denotes the weights of the $k$-th layer of the neural network. The terms $\sigma_{k_\mathcal{P}}$ and $\sigma_{k_\mathcal{Q}}$ represent the variances of the prior and posterior distributions of the weights for $k$-th layer, respectively. These variances reflect the spread of the weight values and are vital to understanding the network's robustness to adversarial attacks; larger variances in the posterior suggest a model more sensitive to input perturbations. These variances are particularly crucial in the adversarial setting as they directly influence the classifier's stability.

The KL divergence is further simplified when the prior and posterior distribution variances are equal, which is a common assumption made to facilitate the calculation. In such cases, we obtain:

$$
\text{KL}[\mathcal{Q}||\mathcal{P}] = \sum_k \frac{\|\mathbf{W}_k\|_{\text{F}}^2}{2\sigma_k^2} + \text{const.}
$$

In this simplified form, the KL divergence is directly proportional to the Frobenius norm of the weight matrices, scaled by the variance of the distributions. It offers a computationally tractable measure for evaluating the divergence in an adversarial machine learning setting. □

## A.3 PROOF OF LEMMA 2.4

*Proof.* The adversarial loss approximation in the context of adversarial ML involves considering the stability of the training loss in the face of adversarial perturbations. Let $K$ be the total number of neural network layers. For any $k \in [K] = \{1, 2, \ldots, K\}$, let $\mathbf{W}_k$ be the $k$-th layer weight matrix of the neural network, $\boldsymbol{w}_k = \text{vec}(\mathbf{W}_k)$ be its vectorized counterpart, and $d_k$ be the dimension of $\boldsymbol{w}_k$. According to our assumptions, we can write $\mathcal{Q} = \mathcal{N}(\theta, \boldsymbol{\Sigma})$ where $\boldsymbol{\Sigma} = [\sigma_1^2\mathbf{I}_{d_1}, \sigma_2^2\mathbf{I}_{d_2}, \ldots, \sigma_K^2\mathbf{I}_{d_K}]$ is the the (diagonal) covariance matrix, where $\sigma_k^2$ denotes the variance of the $k$-th layer $\boldsymbol{w}_k$. Therefore, we can express the difference in empirical losses between the perturbed and unperturbed classifier with parameters $\theta$ as:

$$
\begin{aligned}
\mathbb{E}_{\theta'\sim\mathcal{Q}}[\mathcal{L}_{\mathcal{S}}(\theta',\boldsymbol{x}+\boldsymbol{\delta}(\theta'),y)] &- \mathcal{L}_{\mathcal{S}}(\theta,\boldsymbol{x}+\boldsymbol{\delta}(\theta),y) \\
&= \mathbb{E}_{\Delta\theta\sim\mathcal{N}(\mathbf{0},\boldsymbol{\Sigma})}\big[\mathcal{L}_{\mathcal{S}}(\theta+\Delta\theta,\boldsymbol{x}+\boldsymbol{\delta}(\theta+\Delta\theta),y) - \mathcal{L}_{\mathcal{S}}(\theta,\boldsymbol{x}+\boldsymbol{\delta}(\theta),y)\big],
\end{aligned}
\tag{7}
$$

which captures the averaged adversarial loss over the weight perturbations $\Delta\theta$ drawn from a Gaussian distribution. Since we assume the empirical robust loss can be approximated using the second-

order Taylor expansion around $\theta$, we can simplify Equation 7 as:

$$
\begin{aligned}
\mathbb{E}_{\theta' \sim \mathcal{Q}}&[\mathcal{L}_{\mathcal{S}}(\theta', \boldsymbol{x} + \boldsymbol{\delta}(\theta'), y)] - \mathcal{L}_{\mathcal{S}}(\theta, \boldsymbol{x} + \boldsymbol{\delta}(\theta), y) \\
&\approx \frac{1}{2}\mathbb{E}_{\Delta\theta \sim \mathcal{N}(\boldsymbol{0}, \boldsymbol{\Sigma})}\left[\Delta\theta^{\top}\nabla_{\theta}^{2}\mathcal{L}_{\mathcal{S}}(\theta, \boldsymbol{x} + \boldsymbol{\delta}(\theta), y)\Delta\theta\right] \\
&= \frac{1}{2}\sum_{k \in [K], k' \in [K]}\mathbb{E}_{\Delta\boldsymbol{w}_k \sim \mathcal{N}(\boldsymbol{0}, \sigma_k^2 \mathbf{I}_{d_k})}\mathbb{E}_{\Delta\boldsymbol{w}_{k'} \sim \mathcal{N}(\boldsymbol{0}, \sigma_{k'}^2 \mathbf{I}_{d_{k'}})}\left[\Delta\boldsymbol{w}_k^{\top}\mathbf{H}_{kk'}\Delta\boldsymbol{w}_{k'}\right] \\
&= \frac{1}{2}\sum_{k \in [K]}\mathbb{E}_{\Delta\boldsymbol{w}_k \sim \mathcal{N}(\boldsymbol{0}, \sigma_k^2 \mathbf{I}_{d_k})}\left[\Delta\boldsymbol{w}_k^{\top}\mathbf{H}_{kk}\Delta\boldsymbol{w}_k\right] \\
&= \frac{1}{2}\sum_{k \in [K]}\mathrm{Tr}(\mathbf{H}_{kk}) \cdot \sigma_k^2,
\end{aligned}
\tag{8}
$$

where $\mathbf{H}_{kk'}$ is a $d_k \times d_{k'}$ matrix representing the second-order derivative of the empirical robust loss with respect to the $k$-th layer's vectorized weight parameters $\boldsymbol{w}_k$ and the $k'$-th layer's vectorized weight parameters $\boldsymbol{w}_{k'}$:

$$
\mathbf{H}_{kk'} = \frac{\partial^2 \mathcal{L}_{\mathcal{S}}(\theta, \boldsymbol{x} + \boldsymbol{\delta}(\theta), y)}{\partial \boldsymbol{w}_k \cdot \partial \boldsymbol{w}_{k'}} \quad \text{for any } k \in [K] \text{ and any } k' \in [K].
\tag{9}
$$

For simplicity, we write $\mathbf{H}_k = \mathbf{H}_{kk'}$ which corresponds to the Hessian matrix of the empirical robust loss with respect to the $k$-th layer. Here in Equation 8, the second equality holds because the covariance matrix $\mathrm{cov}(\Delta\boldsymbol{w}_k, \Delta\boldsymbol{w}_{k'}) = \boldsymbol{0}$ (for any $k \neq k'$), and the last equality follows the singular value decomposition of $\mathbf{H}_k$ and the fact that $\Delta\boldsymbol{w}_k$ follows an isotropic Gaussian distribution.

Note that the Hessian matrix $\mathbf{H}_k$ encapsulates the second-order partial derivatives of the loss function with the weights of the layer, indicating how the loss curvature changes in response to perturbations in the weights. Equation 8 suggests that we can approximate the Classifier Variability Component by the trace of the Hessian matrices, which completes the proof of Lemma 2.4. This approximation provides a computationally efficient method to evaluate the classifier's sensitivity to adversarial perturbations, offering insights into the model's robust generalization capabilities. $\square$

## A.4 PROOF OF THEOREM 2.5

*Proof.* According to Lemma 2.3 and Lemma 2.4, we obtain

$$
\begin{aligned}
\lambda^{-1}&\mathrm{KL}[\mathcal{Q}||\mathcal{P}] + \mathbb{E}_{\theta \sim \mathcal{Q}}[\mathcal{L}_{\mathcal{S}}(\theta, \boldsymbol{x} + \boldsymbol{\delta}, y)] - \mathcal{L}_{\mathcal{S}}(\theta, \boldsymbol{x} + \boldsymbol{\delta}, y) \\
&\approx \lambda^{-1}\left(\sum_k \frac{\|\mathbf{W}_k\|_{\mathrm{F}}^2}{2\sigma_k^2} + \mathrm{const}\right) + \frac{1}{2}\sum_k \mathrm{Tr}(\mathbf{H}_k) \cdot \sigma_k^2 \\
&= \frac{1}{2}\sum_k \left(\frac{\|\mathbf{W}_k\|_{\mathrm{F}}^2}{\lambda\sigma_k^2} + \mathrm{Tr}(\mathbf{H}_k) \cdot \sigma_k^2\right) + \mathrm{const} \\
&= \sum_k \sqrt{\frac{\|\mathbf{W}_k\|_{\mathrm{F}}^2}{\lambda\sigma_k^2} \cdot \mathrm{Tr}(\mathbf{H}_k) \cdot \sigma_k^2} + \mathrm{const} \\
&= \frac{1}{\sqrt{\lambda}}\sum_k \sqrt{\|\mathbf{W}_k\|_{\mathrm{F}}^2 \cdot \mathrm{Tr}(\mathbf{H}_k)} + \mathrm{const},
\end{aligned}
$$

where we set the variances $\sigma_k^2 = \sqrt{\frac{\|\mathbf{W}_k\|_{\mathrm{F}}^2}{\lambda\mathrm{Tr}(\mathbf{H}_k)}}$ to keep the bound to be smallest. Thus, we complete the proof of Theorem 2.5. $\square$

## A.5 PROOF OF EQUATION 6

*Proof.* Consider two sequences where each $a_i = \|\mathbf{W}_k\|_{\mathrm{F}}^2$ (the squared Frobenius norm of the weight matrix at layer $l$) and each $b_i = \mathrm{Tr}(\mathbf{H}_k)$ (the trace of the Hessian matrix at the same layer).

Applying the Cauchy-Schwarz inequality to these sequences, we obtain:

$$\left( \sum_k \sqrt{\|\mathbf{W}_k\|_{\mathrm{F}}^2 \cdot \mathrm{Tr}(\mathbf{H}_k)} \right)^2 \leq \left( \sum_k \|\mathbf{W}_k\|_{\mathrm{F}}^2 \right) \left( \sum_k \mathrm{Tr}(\mathbf{H}_k) \right).$$

Here, $\|\mathbf{W}_k\|_{\mathrm{F}} \sqrt{\mathrm{Tr}(\mathbf{H}_k)}$ approximates the square root of $\|\mathbf{W}_k\|_{\mathrm{F}}^2 \cdot \mathrm{Tr}(\mathbf{H}_k)$, reflecting the product of the norm of the weight and the square root of the curvature term's trace. Taking the square root of the inequality, we get:

$$\sum_k \sqrt{\|\mathbf{W}_k\|_{\mathrm{F}}^2 \cdot \mathrm{Tr}(\mathbf{H}_k)} \leq \sqrt{\left( \sum_k \|\mathbf{W}_k\|_{\mathrm{F}}^2 \right)} \cdot \sqrt{\left( \sum_k \mathrm{Tr}(\mathbf{H}_k) \right)},$$

where the left-hand side of the above equality corresponds to our definition of WCI (Definition 2.6). This concludes the proof that WCI is bounded by the geometric mean of the total weight norms squared and the total curvature across all layers. $\qquad\square$

## B   DETAILED EXPERIMENTAL SETTINGS

### B.1   DATASETS AND MODELS

We conduct experiments on the CIFAR-10, CIFAR-100, and SVHN datasets, which are widely used benchmarks for evaluating the robustness of deep learning models. CIFAR-10 and CIFAR-100 consist of $60,000$ $32 \times 32$ color images in 10 and 100 classes, respectively, with $50,000$ training images and $10,000$ test images (Krizhevsky et al., 2009). SVHN contains $32 \times 32$ color images of house numbers, with $73,257$ training images and $26,032$ test images. We use the standard data splits for training and testing the models (Netzer et al., 2011). We use PreActResNet18 (He et al., 2016) as the base architecture for all experiments, a widely used model for adversarial training. We train the models using the standard cross-entropy loss and the SGD optimizer with a learning rate of $0.1$, a weight decay of $5 \times 10^{-4}$, and a momentum of $0.9$. We also apply the piecewise learning rate schedule that reduces the learning rate by a factor of 10 at epochs 100 and 150. We train the models for 200 epochs with a batch size of 128.

### B.2   ADVERSARIAL TRAINING

We perform PGD-based adversarial training (Madry et al., 2017), which is a widely used approach for training robust models. In particular, we consider $\ell_\infty$-norm bounded perturbations with strength $\epsilon = 8/255$. We generate adversarial examples by applying PGD with a step size of $2/255$ (for SVHN $1/255$) and 10 iterations. We use the same adversarial parameters for all experiments to ensure consistency. We train the models using adversarial examples generated during training to improve robustness. We also evaluate the models on adversarial examples generated during testing to assess their robustness against unseen attacks. We use the standard $\ell_\infty$ norm for generating adversarial examples, which is a common choice for evaluating robustness against perturbations.

### B.3   EVALUATION METRICS

We evaluate the models using standard metrics, including robust accuracy, robust error, and robust loss. The robust accuracy is the percentage of correctly classified adversarial examples, while the robust error is the percentage of misclassified ones. The robust loss is the average loss over the adversarial examples. We also calculate the generalization gap, which is the difference between the standard and robust error rates, to quantify the model's generalization performance. We use these metrics to assess the robustness and generalization capabilities of the models across different settings and conditions.

## C   ADDITIONAL EXPERIMENTS

### C.1   CORRELATION ANALYSIS OF WCI AND GENERALIZATION GAP IN SECTION 3

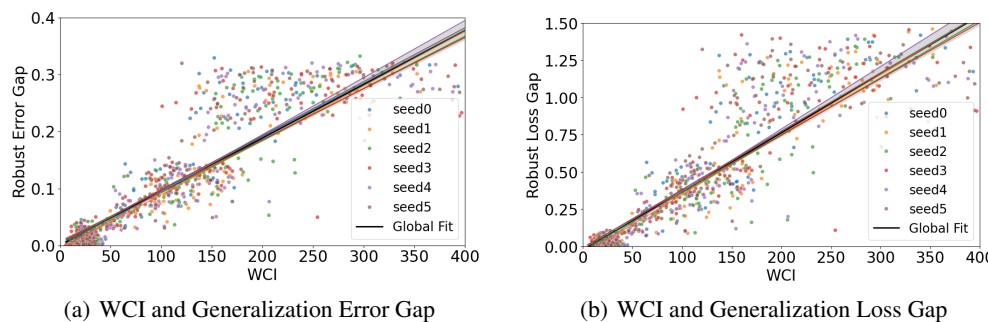

(a) WCI and Generalization Error Gap                    (b) WCI and Generalization Loss Gap

Figure 5: Corralation between WCI and generalization gaps (error and loss) on CIFAR-10. Different colors represent different seeds, and the straight lines represent the linear regression fit.

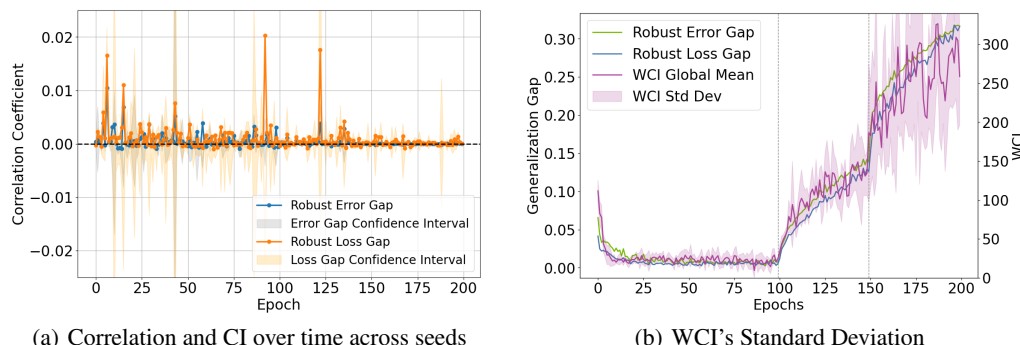

(a) Correlation and CI over time across seeds            (b) WCI's Standard Deviation

Figure 6: Figure (a) shows correlation and confidence interval over time across seeds on CIFAR-10, and Figure (b) shows WCI's standard deviation over time with seeds on CIFAR-10.

The experiments are conducted on CIFAR-10 with 6 different seeds, where we observe that the WCI values are consistent across different seeds, indicating that the WCI is a stable metric for evaluating the robustness of models. We analyze the correlation between the WCI and the generalization gap, which is the difference between the standard and robust error rates. We discover that the WCI values are positively correlated with the generalization gap, indicating that models with higher WCI values tend to have larger generalization gaps. This relationship highlights the importance of the WCI as a reliable indicator of generalization performance in adversarial training.

**Individual Hypothesis Test.** For each individual seed, we conduct a separate hypothesis test to evaluate the linear correlation between the WCI and the gap measures. The null hypothesis (H0) asserts that there is no linear correlation between WCI and the gap measure, mathematically expressed as $H_0 : \rho = 0$, where $\rho$ denotes the population correlation coefficient. Conversely, the alternative hypothesis (H1) posits that there is a statistically significant linear correlation between WCI and the gap measure, expressed as $H_1 : \rho \neq 0$. This testing framework allows us to determine whether the observed correlations are statistically significant for each seed. We use the Pearson correlation coefficient to quantify the strength of the linear relationship between the WCI and the gap measures. The p-values for all seeds are less than $0.05$, indicating that the observed correlations are statistically significant, thereby confirming the strong relationship between WCI and generalization gaps.

**Global Hypothesis Test.** After evaluating individual seeds, we aggregate the data to test the global hypothesis. The global null hypothesis (H0) suggests that the WCI is not consistently correlated with the gap measures across all seeds. Conversely, the global alternative hypothesis (H1) asserts that WCI is consistently and significantly correlated across all seeds.

Following our analysis of individual seeds and the aggregation of data to evaluate the global hypothesis, we have obtained significant results confirming a robust correlation between the WCI and the gap measures. The global regression for the Robust Error Gap yielded a WCI coefficient of $0.0007$ with a 95% confidence interval of $[0.0007, 0.0008]$ and an $R^2$ value of $0.6790$. For the Robust Loss

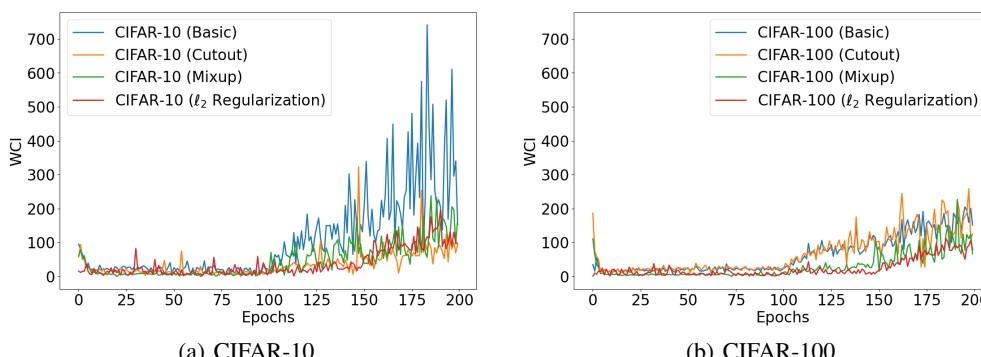

(a) CIFAR-10    (b) CIFAR-100

Figure 7: WCI curves with various regularization techniques on CIFAR-10 and CIFAR-100.

Gap, the WCI coefficient was 0.0030 with a 95% confidence interval of $[0.0029, 0.0032]$ and an $R^2$ of 0.6848. These p-values, $6.8182 \times 10^{-294}$ and $1.2305 \times 10^{-298}$ respectively, strongly reject the global null hypothesis that WCI does not consistently correlate with the gap measures.

The comparison between each seed's correlation coefficients and the global coefficients confirms that each seed's results align with the global trend. As shown in Figure 5, all seeds show their coefficients within the global confidence intervals for both the Robust Error Gap and Robust Loss Gap. This consistent overlap across different training runs again confirms the global alternative hypothesis that WCI is significantly correlated with the gap measures across all seeds.

**Correlation Analysis over Epochs.** Moreover, to assess the robustness of the relationship between the WCI and the generalization gaps, we conducted a statistical correlation analysis across multiple random seeds over time (epoches). As shown on Figure 6, the correlation between the WCI and the generalization gaps basically stay positive and stable over time across different seeds. The confidence intervals for the correlation values are consistent across seeds, further confirming the robustness of the WCI as an indicator of generalization performance. This stability indicates that the WCI is a reliable metric for evaluating the generalization performance of adversarially trained models. Additionally, the standard deviation (Std Dev) for the WCI values combines different seeds still shows an inhibitory trend with the generalization gap, further confirming the robustness of the WCI as an indicator of generalization performance. Note that since we only studied 6 seeds and the WCI fluctuated, the confidence interval is sometimes not very narrow. However, this does not affect our conclusion that WCI is strongly positively correlated with the generation gap.

## C.2  Results for Other Training Algorithms in Section 3

We conduct experiments on both CIFAR-10 and CIFAR-100 datasets with various regularization and data augmentation techniques, including basic adversarial training, cutout, mixup, and $\ell_2$ regularization. The results are presented in Figures 7(a) and 7(b), respectively. Across both datasets and all regularization methods, WCI consistently exhibits similar trends, with higher values correlating with increased robustness losses and generalization gaps, especially during periods of overfitting. This trend emphasizes the widespread nature of overfitting and further confirms the strong correlation between the WCI and the generalization gap, demonstrating that models with higher WCI values tend to exhibit larger generalization gaps. This relationship highlights the importance of the WCI as a reliable indicator of generalization performance in adversarial training.

## C.3  Results for Varying Thresholds of Algorithm 1 in Section 4

In Section 4, we examined the relationship between the WCI and the learning rate during adversarial training. We conducted additional experiments to further investigate this relationship by varying the WCI threshold values between 10 and 100. While the results for the threshold of 100 were included in the main text, here we provide a broader analysis across the full range.

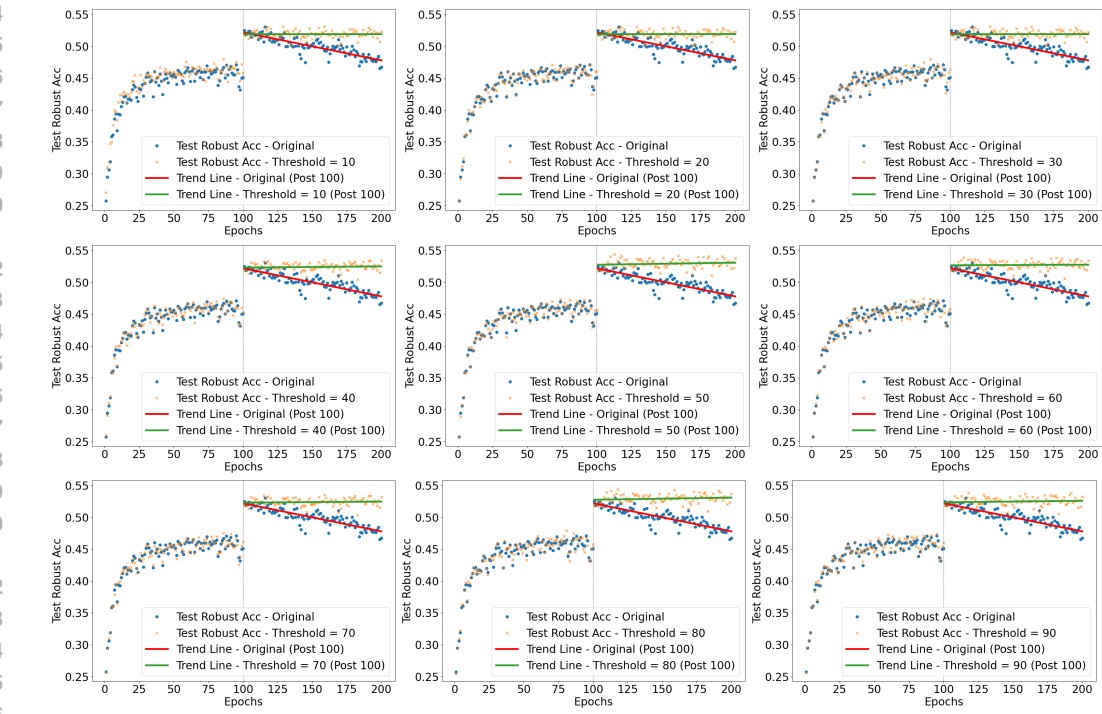

Figure 8: Performance plots for adversarial training with different thresholds used in Algorithm 1.

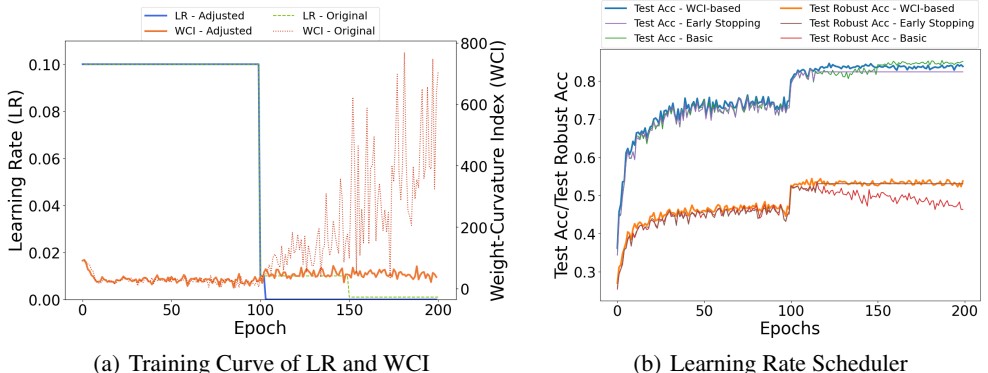

(a) Training Curve of LR and WCI

(b) Learning Rate Scheduler

Figure 9: Analysis of adaptive learning rate and WCI-based adjustments. (a) The dynamics of the learning rate (LR) and Weight-Curvature Index (WCI) during adversarial training. (b) Performance comparison across various learning rate scheduling strategies using standard and robust accuracies.

As shown in Figure 8, our experiments reveal that robust accuracy is not highly sensitive to the exact choice of threshold as long as it falls within a reasonable range. Specifically, thresholds between 10 and 100 consistently lead to effective learning rate adjustments and prevent robust overfitting. This suggests that while the specific threshold value can vary, keeping it within a moderate range ensures optimal performance and robust generalization. These additional results reinforce the flexibility and reliability of the WCI-based approach in dynamically adjusting the learning rate, showing that even with different threshold values, models can maintain both strong robustness and generalization.

## C.4    DETAILED RESULTS FOR UNDERSTANDING WCI DYNAMIC IN SECTION 4

To better understand the experiment in Section 4, we analyze the dynamics of learning rate adjustments and the behavior of the Weight-Curvature Index (WCI) before and after tuning. As depicted in Figure 9(a), the learning rate is dynamically decayed to ensure that the WCI remains stable, indi-

Table 2: Training and testing time with and without WCI.

| Dataset | Training Time ($s$) | | Testing Time ($s$) | |
|---|---|---|---|---|
| | Without WCI | With WCI | Without WCI | With WCI |
| CIFAR-10 | 107.84 | 699.30 | 19.60 | 19.24 |
| CIFAR-100 | 66.23 | 466.14 | 12.48 | 12.49 |
| SVHN | 95.74 | 652.02 | 31.75 | 31.95 |
| CIFAR-10 (test with AutoAttack) | – | 717.35 | – | 1678.7 |

cating an adaptive approach to maintain model stability while training progresses. This adaptation is crucial for balancing the exploration and exploitation phases during training, potentially reducing the risk of overfitting by aligning the learning rate with the underlying model complexity measured by the WCI. Moreover, as shown in Figure 9(b), our comparison between different training strategies highlights significant findings. The WCI-based adjustment not only enhances the standard test accuracy but also improves the testing robust accuracy compared to the basic and early stopping methods. This suggests that incorporating WCI into the training process allows for a more nuanced control over model updates, which is particularly beneficial for adversarial training scenarios where robustness is as critical as accuracy.

## D    COMPUTATIONAL COMPLEXITY ANALYSIS

In this section, we provide a detailed analysis of the computational complexity of the Weight-Curvature Index (WCI) and its impact on training and testing times.

Table 2 presents the training and inference time for adversarial training on CIFAR-10, CIFAR-100, and SVHN datasets with and without using WCI-based learning rate adjustment schemes. In particular, we use 4×NVIDIA A100 40GB Tensor Core GPUs (SXM4 Cards) for training and testing the models. The results show that the training time increases significantly when using WCI due to the additional computation required to calculate the WCI values. However, the testing time remains relatively stable with and without WCI, indicating that the WCI does not significantly impact the inference time. This analysis demonstrates that while the WCI introduces additional computational overhead during training, it does not affect the model's inference performance, making it a practical and efficient method for improving model robustness. It is worth noting that in the proposed dynamic learning rate adjustment strategy (Algorithm 1), WCI is only computed after 100 epochs of adversarial training, which avoids the computational overhead in the initial epochs. We expect the overall training time can be significantly reduced by implementing interval-based WCI adjustments, wherein WCI is computed every few epochs instead of every single epoch post the initial period while largely keeping the benefits of improving robust generalization. Future studies can further explore how to lower the computational costs for calculating the trace of Hessian matrices for WCI by applying Hessian approximation techniques or employing probabilistic methods.

