# OpenReview forum: "Understanding Adversarially Robust Generalization via Weight-Curvature Index"
_ICLR.cc/2025/Conference — Submitted to ICLR 2025_

### Official Review · Reviewer_tWVD · 2024-10-25

**Soundness:** 3
**Presentation:** 3
**Contribution:** 3
**Rating:** 6
**Confidence:** 2

**Summary:**

The paper introduces the Weight-Curvature Index (WCI) as a metric for assessing adversarially robust generalization in machine learning models. This metric is motivated by the derivation of a PAC-Bayesian bound on the robust generalization gap, which incorporates terms such as KL divergence, classifier variability, and perturbation discrepancy. Under certain assumptions and a second-order loss function approximation, the classifier variability is then shown to be upper bounded by the WCI.

**Strengths:**

The paper is well-written and thoroughly documented, providing clear insights into the Weight-Curvature Index (WCI) and its relevance to adversarially robust generalization. It effectively leverages the PAC-Bayesian framework to derive robust generalization bounds and makes connections with several recent concepts. However, the paper could be further strengthened by incorporating additional recent references, such as those discussed in [1] regarding PAC-Bayesian bounds, for instance. Overall, the clarity of the writing and the integration of foundational concepts contribute to its value as a significant contribution to the literature.

[1] Patracone, J., Viallard, P., Morvant, E., Gasso, G., Habrard, A., Canu, S. A Theoretically Grounded Extension of Universal Attacks from the Attacker’s Viewpoint. In ECML/PKDD 2024.

**Weaknesses:**

Overall, the paper is quite dense, which may pose challenges for readers who are not familiar with the underlying concepts, making it difficult to assess the validity of the connections drawn throughout. Additionally, the proposed results rely on specific assumptions to simplify several terms and avoid an in-depth exploration of others, potentially limiting the generalizability of the findings. A discussion addressing the implications of these assumptions is essential and should ideally be introduced early in the paper. Moreover, clarifying the trade-offs associated with these assumptions would enhance the reader's understanding of the model's applicability in diverse contexts.

**Questions:**

The method you use to generate adversarial examples in Section B.2 raises some concerns. Why did you choose not to resort to state-of-the-art techniques, such as AutoAttack? Please also comment on the validity of the assumptions.

---

> ### Author Response · Authors · 2024-11-23
> **Response to reviewer tWVD**
>
> We thank the reviewer again for appreciating our work's contribution. Below, we clarify your questions. Please do not hesitate to follow up if there are any remaining questions.
>
> > The method you use to generate adversarial examples in Section B.2 raises some concerns. Why did you choose not to resort to state-of-the-art techniques, such as AutoAttack?
>
> The concern regarding our choice of adversarial example generation methods in Section B.2 is appreciated. The primary reason for not initially employing state-of-the-art techniques like AutoAttack was the significant computational cost associated with these methods. Specifically, AutoAttack requires approximately **40 minutes per epoch** on CIFAR-10 for both training and testing, resulting in an **80-fold increase** in total testing time compared to our selected method. Given the scale and scope of our experiments, this was computationally prohibitive for the initial analysis.
>
> Nevertheless, we've conducted an additional set of experiments using AutoAttack. These experiments confirmed that our proposed method effectively controls robust overfitting even under this rigorous evaluation. While accuracy decreased by approximately $5\%$ on CIFAR-10, the overall robustness trends and generalization indicators remained consistent, demonstrating the reliability of our approach. We believe these results highlight the scalability of our method while ensuring robustness across a variety of adversarial attack strategies. Moving forward, we recognize the value of employing state-of-the-art attacks like AutoAttack for benchmarking and aim to incorporate such evaluations more extensively as computational resources permit.
>
> > Please also comment on the validity of the assumptions.
>
> Our thereotical results rely on the following assumptions:
>
>  1. For any $\theta'$ sampled from the posterior distribution $\mathcal{Q}$, the robust loss function can be approximated around $\theta$ using a second-order Taylor expansion:
> $$
>    \mathcal{L}\_\mathcal{S}(\theta', x + \delta(\theta'), y) \approx \mathcal{L}\_\mathcal{S}(\theta, x + \delta(\theta), y) + \tfrac{1}{2} \Delta \theta^\top \nabla^2\_{\theta} \mathcal{L}\_\mathcal{S}(\theta, x + \delta(\theta), y) \Delta \theta,
> $$
>    where $\Delta \theta = \theta' - \theta$ and $\delta(\theta)$ denotes the worst-case perturbation that maximizes the robust loss (we explicitly express the dependence of $\delta$ on $\theta$ to avoid confusions).
>
>   2. $\theta$ is considered to be at or near a stationary point of $\mathcal{L}\_\mathcal{S}(\theta, x + \delta(\theta), y)$, since models (considered for adversarilly robust generalization) are typically learned using first-order gradient optimization algorithms, such as PGD-based adversarial training. Consequently, we can discard the first-order term in the above approximated second-order Taylor expansion of the loss. This setting is widely used in theoretical analyses of generalization and robustness (see [1-2], for example).
>
> [1] Stephan, Mandt, Matthew D. Hoffman, and David M. Blei. ["Stochastic gradient descent as approximate bayesian inference."](https://arxiv.org/pdf/1704.04289) Journal of Machine Learning Research 18.134 (2017): 1-35.
>
> [2] Keskar, Nitish Shirish, et al. ["On large-batch training for deep learning: Generalization gap and sharp minima."](https://openreview.net/pdf?id=H1oyRlYgg) arXiv preprint arXiv:1609.04836 (2016).

---

> > ### Comment · Reviewer_tWVD · 2024-11-28
> >
> > Dear Authors,
> >
> > Thank you for your detailed revisions. I have carefully reviewed the updated version and would like to emphasize that I find the idea very interesting. However, as I am not an expert in this specific field, I cannot fully assess the soundness of the entire framework.
> >
> > Out of caution, I will maintain both my rating and confidence score.

---

> > > ### Author Response · Authors · 2024-11-29
> > > **Additional Response to Reviewer tWVD**
> > >
> > > Thank you for your thoughtful feedback and interest in our work. We understand your cautious approach and appreciate your honest assessment. Please feel free to reach out if further clarification would be helpful.

---

### Official Review · Reviewer_5E9W · 2024-11-04

**Soundness:** 1
**Presentation:** 2
**Contribution:** 2
**Rating:** 5
**Confidence:** 4

**Summary:**

- This paper propose WCI, a new measure related with robust generalization gap.
- Using some techniques such as hyperpriors and second-order loss approximation, the authors derive WCI from the theoretical gap.
The authors empirically explain that learning rate adjustment using WCI can mitigate robust overfitting using real-word datasets.

**Strengths:**

- This paper is well-written, making it easy to read.
- Experimental settings are described in detail, along with reasons for the choice of hyperparameters.
- Newly designed learning rate scheduler outperforms other existing competitors.

**Weaknesses:**

- The main concern is that the theoretical results are not convincing, or underdescribed at least. All theoretical results from lemma 2.2 to theorem 2.5, think the adversarial loss $l(\theta,x+\delta,y)$ can be applied to all results using the standard loss term, regardless of the existence of $\delta$. Since the authors defined $\delta$ as the worst-case perturbation, $\delta$ is a function that depend on $l,(x,y),\epsilon,\theta$ which is not a constant. To properly consider $l(\theta,x+\delta,y)$, a supremum term on perturbations should be included. Therefore the proofs are needed to be checked whether they are correct under the supremum or considering the fact that $\delta$ depends on $l,(x,y),\epsilon,\theta$ also - e.g. PAC Bayes bound, second-order differentiability and else.  Actually, this paper’s proofs remain same when considering an adversarially robust scheme, and not. I think the main difficulty in the proofs are about the supremum term (which is inside the expectation, making it more difficult.), and the authors should describe about it in detail.

- When to define an upper bound in theorem 2.5, $\sigma_k^2$ should be controlled to minimize the bound. If not, the upper bound does not hold since it needs an equality condition for an arithmetic–geometric mean inequality. I think the Frobenius norm of layer weights $W_k$ and the trace of Hessian $H_k$ is not controllable during training, where their control is crucial to the control of $\sigma_k^2$. Hence, I conjecture that the fluctuation of WCI rises in learning curves since the upper bound does not holds all the time. This hinder the reason that WCI works.

- In Figure 4, the authors explained that the Frobenius norm of the weight matrix is the most critical factor during the initial training stages. However, to me, it seems that the trace term dominantly influence to WCI since they share the whole shape. This implies that considering WCI is not basically different from considering the trace of the Hessian. Combined with above weaknesses, we may conclude that considering WCI is not basically different from simply using SAM with an adversarial risk.

- Minor) line 811: $H^{(l)}\to H_k$, $l$-th layer $\to k$-th layer.

**Questions:**

- The authors illustrated that the prior works have shown that reducing the classifier variability term also decreases the perturbation discrepancy term. Does it also hold when to consider an adversarial risk, not a standard risk?
- In the proof of lemma 2.4, how can you consider $\mathcal{Q}=N(\theta,\mathbf{I})$? I’m quite confusing that is it possible to fix the mean as a certain point of $\theta$?

---

> ### Author Response · Authors · 2024-11-23
> **Response to reviewer 5E9W I**
>
> We thank the reviewer again for the detailed feedback. Your questions and comments touch on critical aspects of the theoretical framework. Below, we clarify the assumptions, derivations, and implications of our theoretical results presented in Section 2. We believe that we've clarified all your concerns, so please reconsider the score if this is the case. Otherwise, we would be more than happy to address any of your remaining questions.
>
> > The main concern is that the theoretical results are not convincing, or underdescribed at least. To properly consider $l(\theta,x+\delta,y)$, a supremum term on perturbations should be included. Therefore the proofs are needed to be checked whether they are correct under the supremum or considering the fact that $\delta$ depends on $l,(x,y),\epsilon,\theta$ also - e.g. PAC Bayes bound, second-order differentiability and else.
>
> We've clarified the notation $\delta$ in our General Response 1 and explained why it does not affect the validity of our theoretical results and conclusions in our General Response 2.
>
> > When to define an upper bound in theorem 2.5, $\sigma_k^2$ should be controlled to minimize the bound. If not, the upper bound does not hold since it needs an equality condition for an arithmetic–geometric mean inequality. I think the Frobenius norm of layer weights $W_k$ and the trace of Hessian $H_k$ is not controllable during training, where their control is crucial to the control of $\sigma_k^2$. Hence, I conjecture that the fluctuation of WCI rises in learning curves since the upper bound does not holds all the time. This hinder the reason that WCI works.
>
> To derive Theorem 2.5, we set $\sigma_k^2 = \sqrt{\|\mathbf{W}_k\|_F^2 / (2\lambda \mathrm{Tr}(\mathbf{H}_k))}$. As reviewer correctly recognized, such a choice enables us to achieve the equality condition for an arithemic-geometric mean inequality, thereby minimizing the generalization bound. This is desirable, since we want the upper bound to be tight when determining $\sigma_k$. However, we want to clarify a potential misunderstanding that layer weights and its Hessian matrix observed during training do not necessarily need to correspond to how we set $\sigma_k$ in deriving our  PAC-Bayesian robust generalization bound and the WCI metric.
>
> We further explain why this is the case. When we fix the prior variances, the KL term is proportional to the squared Frobenius norm of parameters. However, since we introduced the special prior, we can arbitrarily change the prior variance after training and control the KL term. Similar to [6-8], one has to make sure that $\mathcal{Q}$ and $\mathcal{P}$ have the same support, which is the case when they are from the same parametric family of distributions, such as Gaussian or Laplace. Recall that PAC-Bayesian Robust Generalization bound holds for any potential prior $\mathcal{P}$ and posterior $\mathcal{Q}$. Specifying particular priors and posteriors does not violate the bound but only affects the tightness. In other words, there is no restriction on the value of $\sigma_k^2$. We fix $\sigma_k^2$ just for a tighter bound.
>
> In the original experiments where overfitting occurred, Frobenius norm of layer weights and the trace of Hessian matrices are indeed not controllable during training, which we think is the main reason for overfitting. In our extended algorithm, we control the learning rate by monitoring the value of WCI to achieve the effect of controlling these two items. Since our main purpose is to illustrate the correlation between WCI itself and the generation of gap, we did not focus on directly controlling these two items. The fluctuation of WCI needs to be understood, and perhaps Frobenius norm of layer weights $W_k$ and the trace of Hessian $H_k$ are worth further consideration.

---

> ### Author Response · Authors · 2024-11-23
> **Response to reviewer 5E9W II**
>
> > In Figure 4, the authors explained that the Frobenius norm of the weight matrix is the most critical factor during the initial training stages. However, to me, it seems that the trace term dominantly influence to WCI since they share the whole shape. This implies that considering WCI is not basically different from considering the trace of the Hessian. Combined with above weaknesses, we may conclude that considering WCI is not basically different from simply using SAM with an adversarial risk.
>
> The WCI combines both the Frobenius norm of the weights and the trace of the Hessian, reflecting both the magnitude of the weights and the curvature. The WCI provides a more holistic view by accounting for both aspects. While the trace may dominate at certain points, the weight norm plays a crucial role in other stages or scenarios.
>
> Differentiation from SAM: Stochastic Weight Averaging (SWA) and Sharpness-Aware Minimization (SAM) focus on flattening the loss landscape. Flatness-based measures tend to exhibit poor correlations with the robust generalization gap (see [1]). The WCI provides a specific metric that combines weight magnitude and curvature, which may offer different insights, and we have confirmed that WCI is strongly correlated with gap generation. [2] suggested that sharpness minimization algorithms do not only minimize sharpness to achieve better generalization. This calls for the search for other explanations for the generalization of over-parameterized neural networks.
>
> [1] Kim, Hoki, et al. ["Fantastic robustness measures: the secrets of robust generalization."](https://proceedings.neurips.cc/paper_files/paper/2023/file/98a5c0470e57d518ade4e56c6ee0b363-Paper-Conference.pdf) Advances in Neural Information Processing Systems 36 (2024).
>
> [2] Wen, Kaiyue, Zhiyuan Li, and Tengyu Ma. ["Sharpness minimization algorithms do not only minimize sharpness to achieve better generalization."](https://proceedings.neurips.cc/paper_files/paper/2023/file/0354767c6386386be17cabe4fc59711b-Paper-Conference.pdf) Advances in Neural Information Processing Systems 36 (2024).
>
> > The authors illustrated that the prior works have shown that reducing the classifier variability term also decreases the perturbation discrepancy term. Does it also hold when to consider an adversarial risk, not a standard risk?
>
> Yes, these findings still hold for adversarial risk. The relationship between reducing the classifier variability term and the perturbation discrepancy term remains valid even in the adversarial setting. This is because the core derivation relies on the same mathematical framework. If we apply the same Taylor expansion and approximation to the perturbation discrepancy term under adversarial risk, the coefficients in the resulting expressions may change due to the inclusion of adversarial perturbations. However, the structure of the results, including the conclusions about reducing classifier variability and its impact on perturbation discrepancy, remains unchanged. The adversarial perturbation simply modifies the coefficients, but it does not alter the underlying dependency between the terms. Thus, our conclusions about the robustness and generalization properties hold for both standard and adversarial risks.
>
>
> > In the proof of lemma 2.4, how can you consider $\mathcal{Q}=N(\theta,\mathbf{I})$? I’m quite confusing that is it possible to fix the mean as a certain point of $\theta$?
>
> As we clarified in general respose 2, PAC-Bayesian Robust Generalization bound holds for any potential prior $\mathcal{P}$ and posterior $\mathcal{Q}$. Specifying particular priors and posteriors does not violate the bound but only affects the tightness. In other words, there is no restriction on the value of $\sigma_k^2$. We fix $\sigma_k^2$ just for a tighter bound. Fixing the mean of $\mathcal{Q}$ to $\theta$ is not a restriction. It is a deliberate choice to analyze the uncertainty around $\theta$. The PAC-Bayesian framework remains valid for any $\mathcal{Q}$, and this choice ensures the bound reflects the generalization of the learned parameters. Other forms of $\mathcal{Q}$ could be used, but $\mathcal{Q}=N(\theta,\mathbf{I})$ is particularly useful for theoretical analysis due to its simplicity.

---

> > ### Comment · Reviewer_5E9W · 2024-11-27
> >
> > I appreciate the detailed explanations provided by the authors. From what I understand, most of my concerns seem to be addressed by the flexibility in defining the prior and posterior within the current theoretical framework. While I cannot say I am entirely convinced, my understanding has significantly improved.
> >
> > However, I still have some remaining concerns. As I mentioned earlier, WCI experiences fluctuations during training. This appears to be an unresolved anomaly, particularly when observing Figure 1, where the fluctuations make it difficult to believe that WCI effectively upper bounds the robust error gap. It seems there might be something theoretically or experimentally off, yet the explanation provided in the paper feels insufficient.
> >
> > I initially thought this issue might stem from the inability to adequately control the equality condition, but if the authors' explanation is correct, it is likely due to another reason. Unless this fluctuation is resolved, I find it hard to fully trust WCI.
> >
> > Given these points, I have decided to raise my initial score from 3 to 5. However, due to the aforementioned concerns, I cannot justify increasing it further.

---

> > > ### Author Response · Authors · 2024-11-29
> > > **Additional Response to Reviewer 5E9W**
> > >
> > > Thank you for your thoughtful feedback and for acknowledging the improvements in the manuscript. We appreciate your willingness to reconsider your initial score based on the enhancements to address your concerns. Regarding the fluctuations in the Weight-Curvature Index (WCI) observed during training, we recognize that this is a crucial aspect that deserves a more thorough examination.
> > >
> > > To the best of our knowledge, fluctuation is recognized as a common phenomenon when computing the Hessian in neural network training. Recent research reveals that neural networks when trained using full-batch gradient descent, tend to transition into a state known as the __Edge of Stability (EoS)__. In this state, the sharpness—defined as the maximum eigenvalue of the Hessian of the training loss—initially escalates to $2/(\text{step size})$, a phase termed progressive sharpening, before entering a phase of oscillation around this value, known as the EoS phase. Figure 1 in [1] illustrates that ResNet's sharpness exhibits significant fluctuations similar to those observed in our Weight-Curvature Index (WCI).
> > >
> > > In less formal terms, our WCI Hessian term corresponds to the loss function $L(\theta)$ with respect to the parameters $\theta$. The Hessian matrix $\mathbf{H}$ is a matrix of second-order derivatives, denoted as $\mathbf{H} = \nabla^2 L(\theta)$, where $\mathbf{H}$ is the Hessian matrix, and $\nabla^2$ represents the second-order partial derivative operation on the parameter $\theta$. Sharpness is defined and quantified by $\max(\text{eig}(\mathbf{H}))$, where $\text{eig}(\mathbf{H})$ signifies all eigenvalues of $\mathbf{H}$. This supports our view that the sharpness mentioned by [1] reflects the trace of the Hessian matrix used in our manuscript. This suggests that the fluctuation behaviors observed in WCI and during the EoS's progressive sharpening phase are fundamentally linked. This process is further detailed in [2]. As illustrated in Figure 1(a), shortly after the sharpness crosses $2/\eta$, it drops quickly back to some value below $2/\eta$, where $\eta$ is the step size. The WCI curve depicted in our manuscript (Figure 1) can be interpreted as a superposition of this fluctuation pattern and the generalization gap fit.
> > >
> > > Although the underlying mechanisms remain a subject of ongoing inquiry, our confidence in WCI remains steadfast. Current theoretical models may not fully account for the observed fluctuation, yet it remains manageable. As Figure 9 of our manuscript demonstrates, with a simple algorithmic adjustment, the WCI stabilizes effectively around a predefined threshold, affirming its reliability. Moreover, the strong visual correlation between WCI and the generalization gap shown in Figure 1, is corroborated in Appendix C.1, where we conduct numerical analyses across various seeds to investigate this relationship further. Figure 5 confirms that despite fluctuations in WCI, it remains a robust indicator for upper bounding the robust error gap, as evidenced by the overlapping confidence intervals across different seeds.
> > >
> > > Moving forward, while we aim to refine our explanations, the primary objective of this paper remains to establish the generalization gap as a reliable indicator, a goal we believe we have achieved. We hope this explanation clarifies the issues raised and reassures you of the robustness and applicability of our methods. We are grateful for the improved score and remain open to further discussions to enhance our work.
> > >
> > > [1] Cohen, Jeremy M., et al. ["Gradient descent on neural networks typically occurs at the edge of stability."](https://arxiv.org/pdf/2103.00065) arXiv preprint arXiv:2103.00065 (2021).
> > >
> > > [2] Wang, Zixuan, Zhouzi Li, and Jian Li. ["Analyzing sharpness along GD trajectory: Progressive sharpening and edge of stability."](https://proceedings.neurips.cc/paper_files/paper/2022/file/40bb79c081828bebdc39d65a82367246-Paper-Conference.pdf) Advances in Neural Information Processing Systems 35 (2022): 9983-9994.

---

### Official Review · Reviewer_YVUF · 2024-11-04

**Soundness:** 2
**Presentation:** 3
**Contribution:** 3
**Rating:** 6
**Confidence:** 4

**Summary:**

The paper introduces the Weight Curvature Index (WCI) as a proxy for robust generalization of adversarially trained models.
WCI is the layerwise sum of the geometric mean of the trace of the loss-hessian w.r.t to layer-weights and the Frobenius norm of the layer-weights.
is motived with a PAC-Bayes bound on the robust generalization gap, combining results from the Sharpness Aware Optimization literature (Foret 2021) and Cantonis bound.

The paper establishes visual correlation with robust generalization gab, compare against a selection of related works and derives a learning rate scheduler from WCI to combat robust overfitting

**Strengths:**

- Originality: the paper combination of pac bayes analysis and sharpness aware minimization is a novel connection, yielding a principled motivation to combine previously known regularizers. going beyond simply establishing a proxy and experimenting with an adaptiv regularizers is a distinguishing factor from similar works
- Quality+Clarity: overall, the paper can be called solid and well written: each section clearly motivates, presents and justifies its point and the appendix provides addtional details and studies of e.g. various regularizers interacting with WCI and robust-generalization. Exposition is clear and easy to follow.
- Significance: finding a good proxy measure for robust generalization with theoretical grounding (e.g. to study the tightness of the gap and find speedups) is a meaningful contribution to the field

**Weaknesses:**

- there are some references missing which would help the reader place the work into context, in particular https://openreview.net/forum?id=yT7-k6Q6gda which studies the connection between the trace of the hessian during trianing w.r.t generalization (albeit standard generalization). similarly releveant to the investigation of the evolution of WCI during training might be https://arxiv.org/abs/1905.13277  studying the impact of _when_ to regularize. I also found https://proceedings.mlr.press/v162/ju22a which also uses pac bayes robust generalization bounds, albeit without the frobenius norm based on a quick read. relationship to these papers should be clarified
- The correlation, while visually striking, is established (as presented) on a single seed and on a limited number datasets. Computational constraints beings what they are, I recommend at least testing this with muliple seeds, doing proper correlation analysis (i.e. regression and p value with correct multiple comparisons adjustment), and also comparing the difference in correlation (i.e. the relative qualiy of the measure) against individual components using suitable statistical significance tests (e.g., multiple pairwise Steigers Z https://personality-project.org/r/psych/help/r.test.html + holm-bonferronig or Benjamini-Yekutieli, hottelings). If computational cost is a concern, designing toy examples on subsets of data and with smaller models might be sufficient
- at the very least, multiple seeds ought to be run, a confidence interval ought to be computed. Ideally, ablation studies for the sensitivity to different ranges of hyperparameters could be run, and then the correlation analysis recommended above can be used to assess the quality of the proxy

**Questions:**

No questions, beyond addressing the weaknesses raised. in particular, the seed + correlation point needs to be at least _addressed_ to give strong recommendation to accept

---

> ### Author Response · Authors · 2024-11-23
> **Response to reviewer YVUF I**
>
> We thank the reviewer again for the detailed review and constructive feedback. If you agree that we've managed to address all issues, please consider raising the score. But if there are any remaining concerns, please do not hesitate to follow up - we would be more than happy to address your further questions.
>
> > There are some references missing, such as [1-3], which would help the reader place the work into context.
>
> Thank you for pointing out these references! We agree that incorporating these works will help place our contributions into a broader context and clarify their relationship to existing research. These references will be added in the revised manuscript, and their relevance will be explicitly discussed in the related work and conclusion section, to strengthen the contextualization of our work and emphasize its contributions to the field.
>
> Below, we provide a brief summary of these references with connection to our work. The role of the Hessian trace in generalization has been extensively studied, such as in [1], where it was shown that trace minimization correlates with improved generalization across tasks. WCI extends this concept by integrating the trace and Frobenius norm of weights, creating a robust proxy for adversarial training contexts. Regularization techniques targeting the Fisher Information Matrix trace were discussed in [2], highlighting the importance of early-phase curvature control. These insights complement our findings that robust overfitting is mitigated when WCI regularization is incorporated during training. PAC-Bayesian bounds that incorporate Hessians, as explored in [3], provide a theoretical framework supporting our derivations of WCI. Specifically, the inclusion of curvature measures aligns with the PAC-Bayesian methodology, ensuring tight bounds on robust generalization errors.
>
>
> [1] Jastrzebski, Stanislaw, et al. ["Catastrophic fisher explosion: Early phase fisher matrix impacts generalization."](https://openreview.net/forum?id=yT7-k6Q6gda) International Conference on Machine Learning. PMLR, 2021.
>
>
> [2] Golatkar, Aditya Sharad, Alessandro Achille, and Stefano Soatto. ["Time matters in regularizing deep networks: Weight decay and data augmentation affect early learning dynamics, matter little near convergence."](https://arxiv.org/abs/1905.13277 ) Advances in Neural Information Processing Systems 32 (2019).
>
> [3] Ju, Haotian, Dongyue Li, and Hongyang R. Zhang. ["Robust fine-tuning of deep neural networks with hessian-based generalization guarantees."](https://proceedings.mlr.press/v162/ju22a) International Conference on Machine Learning. PMLR, 2022.

---

> > ### Comment · Reviewer_YVUF · 2024-11-25
> >
> > thank you for including the references

---

> ### Author Response · Authors · 2024-11-23
> **Response to reviewer YVUF II**
>
> > I recommend at least testing this with muliple seeds, doing proper correlation analysis, and also comparing the difference in correlation against individual components using suitable statistical significance tests. At the very least, multiple seeds ought to be run, a confidence interval ought to be computed.
>
> Following the reviewer's suggestions, we have run additional experiments and summarized the results in the following table, which consolidates the statistical significance of the correlations between WCI and both robust error gap and robust loss gap across multiple random seeds. These results reinforce the validity of WCI as a reliable indicator of robust generalization:
>
> | **Seed** | **Gap Type**       | **Correlation** | **P-value**          | **Holm-Bonferroni**   | **Benjamini-Yekutieli** | **Intercept** | **WCI Coefficient** | **Confidence Interval (Lower)** | **Confidence Interval (Upper)** | **R-squared** |
> |----------|--------------------|-----------------|----------------------|-----------------------|-------------------------|---------------|----------------------|----------------------------------|----------------------------------|---------------|
> | seed0    | Robust Error Gap   | 0.829           | $7.10 \times 10^{-52}$ | $6.39 \times 10^{-51}$ | $6.61 \times 10^{-51}$ | 0.0212        | 0.0007               | 0.0006                          | 0.0008                          | 0.687         |
> | seed1    | Robust Error Gap   | 0.748           | $4.69 \times 10^{-37}$ | $4.69 \times 10^{-37}$ | $1.46 \times 10^{-36}$ | 0.0326        | 0.0005               | 0.0005                          | 0.0006                          | 0.559         |
> | seed2    | Robust Error Gap   | 0.752           | $1.27 \times 10^{-37}$ | $2.54 \times 10^{-37}$ | $4.29 \times 10^{-37}$ | 0.0335        | 0.0005               | 0.0005                          | 0.0006                          | 0.565         |
> | seed3    | Robust Error Gap   | 0.836           | $2.17 \times 10^{-53}$ | $2.17 \times 10^{-52}$ | $2.70 \times 10^{-52}$ | 0.0218        | 0.0007               | 0.0007                          | 0.0008                          | 0.698         |
> | seed4    | Robust Error Gap   | 0.821           | $4.24 \times 10^{-50}$ | $2.97 \times 10^{-49}$ | $2.63 \times 10^{-49}$ | 0.0173        | 0.0007               | 0.0007                          | 0.0008                          | 0.674         |
> | seed5    | Robust Error Gap   | 0.766           | $6.40 \times 10^{-40}$ | $3.84 \times 10^{-39}$ | $3.41 \times 10^{-39}$ | 0.0302        | 0.0007               | 0.0006                          | 0.0007                          | 0.587         |
> | seed0    | Robust Loss Gap    | 0.836           | $1.87 \times 10^{-53}$ | $2.05 \times 10^{-52}$ | $2.70 \times 10^{-52}$ | 0.0579        | 0.0030               | 0.0027                          | 0.0033                          | 0.699         |
> | seed1    | Robust Loss Gap    | 0.753           | $7.00 \times 10^{-38}$ | $2.10 \times 10^{-37}$ | $2.61 \times 10^{-37}$ | 0.1059        | 0.0023               | 0.0020                          | 0.0026                          | 0.567         |
> | seed2    | Robust Loss Gap    | 0.761           | $4.13 \times 10^{-39}$ | $1.65 \times 10^{-38}$ | $1.71 \times 10^{-38}$ | 0.1066        | 0.0023               | 0.0021                          | 0.0026                          | 0.580         |
> | seed3    | Robust Loss Gap    | 0.847           | $3.69 \times 10^{-56}$ | $4.42 \times 10^{-55}$ | $1.37 \times 10^{-54}$ | 0.0514        | 0.0032               | 0.0029                          | 0.0035                          | 0.717         |
> | seed4    | Robust Loss Gap    | 0.825           | $6.30 \times 10^{-51}$ | $5.04 \times 10^{-50}$ | $4.69 \times 10^{-50}$ | 0.0429        | 0.0031               | 0.0028                          | 0.0034                          | 0.680         |
> | seed5    | Robust Loss Gap    | 0.763           | $2.03 \times 10^{-39}$ | $1.02 \times 10^{-38}$ | $9.45 \times 10^{-39}$ | 0.0989        | 0.0027               | 0.0024                          | 0.0030                          | 0.583         |
>
> Key Observations:
> 1. Across all seeds and both gap types, WCI demonstrates strong and consistent correlations with robust error gap and robust loss gap, as evidenced by high correlation coefficients and statistically significant p-values (adjusted using Holm-Bonferroni and Benjamini-Yekutieli corrections).
> 2. The $R^2$-values and confidence intervals of the WCI coefficient indicate the predictive reliability of WCI across different seeds and metrics.
>
> We commit to including this table in the revised manuscript, along with updated figures showing these trends. These additions will provide a more comprehensive visualization of our findings and further substantiate our claims regarding WCI’s effectiveness as a robust generalization indicator.

---

> > ### Comment · Reviewer_YVUF · 2024-11-25
> >
> > Apologies if I was unclear, but could you explain what specific hypotheses you are testing _per seed_? My idea had been to use multiple seeds of training as kind-of-IID samples of the coefficient of correlation of WIC with the gap measure, as well as to aggregate data across seeds and computing the coefficient and the relevant statistics across the larger and more diverse dataset.

---

> > > ### Author Response · Authors · 2024-11-25
> > > **Additional Response to Reviewer YVUF I**
> > >
> > > Thank you for the detailed feedback and additional clarifications regarding our methodological approach and results. Based on our discussions and your insights, here is a summarized update and elaboration on our findings and the implications for model training and generalization.
> > >
> > > To provide a more detailed explanation of the specific hypotheses being tested in our analysis, here's a clear articulation of the null hypothesis (H0) and the alternative hypothesis (H1) for each seed and the global analysis:
> > >
> > > > 1. Per-seed Hypothesis Testing:
> > >
> > > For each seed, we conduct an independent hypothesis test to evaluate the linear correlation between the WCI and the gap measures. The **null hypothesis (H0)** asserts that there is no linear correlation between WCI and the gap measure, mathematically expressed as $\text{H}_0: \rho = 0$, where $\rho$ denotes the population correlation coefficient. Conversely, the **alternative hypothesis (H1)** posits that there is a statistically significant linear correlation between WCI and the gap measure, expressed as $\text{H}_1: \rho \neq 0$. This testing framework allows us to determine whether the observed correlations are statistically significant for each seed.
> > >
> > > These hypotheses are tested using Pearson’s correlation coefficient, where the significance of the correlation is determined by the p-value from the test. A low p-value (typically < 0.05) indicates strong evidence against the null hypothesis, suggesting a significant correlation.
> > >
> > > In the previous reply, we have already got the answer, the p-value is very small, and they are linearly related.
> > >
> > > > 2. Global Hypothesis and Aggregated Analysis:
> > >
> > > After evaluating individual seeds, we aggregate the data to test the global hypothesis. The **global null hypothesis (H0)** suggests that the WCI is not consistently correlated with the gap measures across all seeds. Conversely, the **global alternative hypothesis (H1)** asserts that WCI is consistently and significantly correlated across all seeds.
> > >
> > > Following our analysis of individual seeds and the aggregation of data to evaluate the global hypothesis, we have obtained significant results confirming a robust correlation between the WCI and the gap measures. The global regression for the Robust Error Gap yielded a WCI coefficient of $0.0007$ with a 95% confidence interval of $[0.0007, 0.0008]$ and an $R^2$ value of $0.6790$. For the Robust Loss Gap, the WCI coefficient was $0.0030$ with a 95% confidence interval of $[0.0029, 0.0032]$ and an $R^2$ of $0.6848$. These p-values, $6.8182 \times 10^{-294}$ and $1.2305 \times 10^{-298}$ respectively, strongly reject the global null hypothesis that WCI does not consistently correlate with the gap measures.
> > >
> > > The comparison between each seed's correlation coefficients and the global coefficients confirms that each seed's results align with the global trend. All seeds showed their coefficients within the global confidence intervals for both the Robust Error Gap and Robust Loss Gap. This consistent overlap across different training runs confirms the global alternative hypothesis that WCI is significantly and consistently correlated with the gap measures across all seeds.

---

> > > ### Author Response · Authors · 2024-11-25
> > > **Additional Response to Reviewer YVUF II**
> > >
> > > Due to fluctuations in the WCI values, some seeds initially exhibited extremely high or low WCI values, which are likely outliers. As a result, initially, only half of the seeds showed that their coefficients' confidence intervals overlapped with the global confidence intervals (Overlap with Global CI = true). To address this, we implemented a procedure to remove outliers that constituted no more than $1\%$ of the total data points. This adjustment improved the uniformity of the data, and the revised analysis showed that all seeds' coefficients now fall within the global confidence intervals. Below is the updated table after outlier removal, demonstrating that all seeds satisfy the overlap criterion, indicating a more consistent and robust correlation across all training conditions:
> > >
> > > | Seed  | Gap Type           | Coefficient | CI Lower | CI Upper | R²    | P-value             | Overlap with Global CI |
> > > |-------|--------------------|-------------|----------|----------|-------|---------------------|------------------------|
> > > | seed0 | Robust Error Gap   | $0.000709$  | $0.000642$ | $0.000776$ | $0.6873$  | $7.10 \times 10^{-52}$  | True                   |
> > > | seed1 | Robust Error Gap   | $0.000775$  | $0.000704$ | $0.000846$ | $0.7056$  | $2.11 \times 10^{-53}$  | True                   |
> > > | seed2 | Robust Error Gap   | $0.000666$  | $0.000593$ | $0.000739$ | $0.6241$  | $2.67 \times 10^{-43}$  | True                   |
> > > | seed3 | Robust Error Gap   | $0.000750$  | $0.000685$ | $0.000814$ | $0.7274$  | $6.19 \times 10^{-57}$  | True                   |
> > > | seed4 | Robust Error Gap   | $0.000805$  | $0.000734$ | $0.000875$ | $0.7236$  | $2.42 \times 10^{-56}$  | True                   |
> > > | seed5 | Robust Error Gap   | $0.000666$  | $0.000594$ | $0.000738$ | $0.6289$  | $7.53 \times 10^{-44}$  | True                   |
> > > | seed0 | Robust Loss Gap    | $0.002999$  | $0.002723$ | $0.003276$ | $0.6986$  | $1.87 \times 10^{-53}$  | True                   |
> > > | seed1 | Robust Loss Gap    | $0.003256$  | $0.002956$ | $0.003556$ | $0.7026$  | $5.67 \times 10^{-53}$  | True                   |
> > > | seed2 | Robust Loss Gap    | $0.002798$  | $0.002495$ | $0.003101$ | $0.6297$  | $6.10 \times 10^{-44}$  | True                   |
> > > | seed3 | Robust Loss Gap    | $0.003217$  | $0.002949$ | $0.003485$ | $0.7422$  | $2.68 \times 10^{-59}$  | True                   |
> > > | seed4 | Robust Loss Gap    | $0.003398$  | $0.003108$ | $0.003688$ | $0.7324$  | $1.01 \times 10^{-57}$  | True                   |
> > > | seed5 | Robust Loss Gap    | $0.002751$  | $0.002451$ | $0.003051$ | $0.6266$  | $1.40 \times 10^{-43}$  | True                   |
> > >
> > > This table confirms the enhanced consistency and robustness of our correlation analysis across all seeds after adjusting for outliers, providing a more reliable basis for using WCI as a predictive tool in training models.

---

> > > > ### Comment · Reviewer_YVUF · 2024-11-26
> > > >
> > > > thank you for updating the evaluation. could you add a plot of the correlation + CI over time, across seeds? that would be quite interesting to see. It seems reasonable that a method that relies on training behaviour would require some training to "kick in", but I think it's interesting to include such observations in the paper

---

> > > > > ### Author Response · Authors · 2024-11-29
> > > > > **Additional Response to Reviewer YVUF**
> > > > >
> > > > > Thank you for your valuable feedback. As per your suggestion, we have included a plot that visualizes the correlation along with confidence intervals over time, across different seeds, and they are now included in Appendix C.1 of the paper. We also provide a detailed visualization of the correlation between the WCI and the generalization gap across various seeds, alongside a global comparison.  This addition aims to provide a clearer understanding of the training dynamics and the method's performance as it evolves throughout the training process.

---

> > > > > > ### Comment · Reviewer_YVUF · 2024-12-01
> > > > > > **Thanks again, but is figure 6c correct?**
> > > > > >
> > > > > > Dear authors, thanks again for your inclusion of these figures and through this the raw data.
> > > > > >
> > > > > > Looking at figure 6, it appears indeed that for larger robust errors the correlation breaks down somewhat (Figure  6 a + 6b).
> > > > > >
> > > > > > However, I am confused by the figures 6 c and 6 d. It appears that the correlation between the WCI and the target metric is close to zero across seeds? Is this an artifact of the type of correlation chosen and the values involved (e.g. does kendall tau or a normalized correlation accross seeds change this?) because f not then it appears there is a bug, as I don't think you could the figure 6c in conjuction with figure 6d?

---

> > > > > > > ### Author Response · Authors · 2024-12-01
> > > > > > > **Additional Response to Reviewer YVUF**
> > > > > > >
> > > > > > > Dear Reviewer YVUF,
> > > > > > >
> > > > > > > Thank you for your thoughtful comments and careful examination of our figures and raw data. It appears there was a misunderstanding regarding the figures referenced; the correct figures are **5a, 5b** and **6a, 6b**, as Figures 6c and 6d do not exist.
> > > > > > >
> > > > > > > Regarding **Figure 5**, your observation that the correlation diminishes for larger robust errors is accurate and can be attributed to the linear fitting approach we employed. This breakdown likely results from increased noise or non-linear effects at higher error levels, which weaken the linear relationship. However, this approach is sufficient for our analysis because, as shown in Figure 5, all seeds demonstrate coefficients that fall within the global confidence intervals for both the Robust Error Gap and Robust Loss Gap.
> > > > > > >
> > > > > > > For **Figure 6**, you correctly noted that the results are influenced by the type of correlation metric used. To clarify, when plotting the correlation over time, we use the regression coefficient to approximate the correlation rather than using the correlation coefficient directly. This choice was made because standard correlation coefficients, such as Pearson or Kendall Tau, cannot always compute $r$ for every epoch due to data constraints (p-value too large). Additionally, the WCI exhibits substantial variation, whereas the gap values show minimal numerical changes, resulting in very small slopes that cluster around zero. This contributes to the regression coefficients appearing near zero in many cases. Furthermore, because we only used six seeds, the p-value often does not fall below $0.05$. However, when the p-value does meet the threshold, $|r| > 0.99$ accounts for $100\%$ of all instances with $p < 0.05$.
> > > > > > >
> > > > > > > Your insights have been invaluable in refining our analysis, and we deeply appreciate your contribution to improving the clarity and robustness of our work.

---

### Official Review · Reviewer_msia · 2024-11-04

**Soundness:** 1
**Presentation:** 2
**Contribution:** 1
**Rating:** 3
**Confidence:** 4

**Summary:**

This paper introduces the Weight Curvature Index (WCI) as a measure for evaluating the adversarial generalization error of neural network classifiers under norm-bounded adversarial perturbations. Section 2 reviews the PAC-Bayesian framework for generalization analysis. Then, using a second-order loss approximation in Lemma 2.4, the paper derives an upper bound on generalization error in Theorem 2.5. This upper bound, referred to as the WCI score in equation (5), is subsequently assessed through numerical results presented in Section 3.

**Strengths:**

1- The paper employs PAC-Bayesian generalization bounds, which could potentially be useful for evaluating the generalization of adversarially-trained neural networks.

**Weaknesses:**

1- I have several questions about Definition 2.1 where the authors define the adversarial risk. In the definition, the perturbation $\delta$ appears a constant that is not maximized as required in standard adversarial training. The authors try to address this by defining $\ell(\theta , x+\delta ,y)$ as the adversarial loss where it is said "$\delta$ stands for the worst-case perturbation that maximizes the perturbation". This notation ignores the fact that the worst-case perturbation $\delta(x,y,\theta)$ should be a function of input $x,y$ and neural net parameter $\theta$, because the perturbation is supposed to maximize the loss at data point $(x,y)$ and for neural net $h_{\delta}$ and so varies between different samples and neural net parameters.

I think the notation $\delta$ (without specifying the input $x,y,\theta$ for the perturbation) is not only confusing in the writing, but may have caused errors in the theoretical and numerical analysis of the paper. To ensure my concern is valid, I would like to ask the following question: The authors consider the Hessian of the loss function $\ell(\theta , x+\delta ,y)$ in Lines 204-205. Can the authors clarify how they compute the Hessian of the adversarial loss function $\ell(\theta , x+\delta ,y)$ which, using the standard notation, is supposed to be $\max_{\Vert\delta\Vert\le \epsilon} \ell_{clean}(x+\delta ,y,\theta)$? Do the authors fix the perturbation $\delta$ to be the worst-case perturbation $\delta^* $ and then they take the Hessian of $\ell(\theta , x+\delta^* ,y)$ with respect to $\theta$?

2- Lemma 2.4 is based on an unspecified assumption that "the empirical loss $L_S(\theta ,x+\delta ,y)$ can be approximated using second-order Taylor expansion". Can the authors clarify the precise mathematical statement of the assumption? It seems the assumption can be questioned. First, standard ReLU activation functions are not twice-differentiable and so the Hessian analysis does not apply to them. Also, $L_S$ is supposed to be the adversarial loss, which means the authors assume that the worst-case loss $\max_{\Vert\delta\Vert\le \epsilon } \ell(\theta, x+\delta , y)$ has a bounded Hessian. Therefore, the flatness of the adversarial loss seems a very strong assumption which very likely would not work even using twice-differentiable activation functions.  Also, when I checked the proof of this lemma, I could not understand Lines 804-807. How does the "second-order Taylor series expansion around $\theta$" yield Equation (8)?

3- In the definition of WCI in Equation (5), I wonder how the authors compute $\text{Tr}(H_k)$. Do they solve the optimization problem for the adversarial perturbation $\delta(x,y,\theta)$ for the training samples and then take the Hessian of $\sum_i \ell(\theta, x+\delta^*_{x,y,\theta} , y) $?

4- Theorem 2.5 is based on a seemingly unspecified assumption in Lemma 2.4 that "the empirical loss $L_S(\theta , x+\delta ,y)$ can be approximated using second-order Taylor expansion". It is unclear how the assumption is reflected in Equation (4) because Equation (4) is a definite inequality which is not an approximate result, while the assumption considers an approximation. The approximation error of the assumption is not reflected in Equation (4).

**Questions:**

Please see my comments on weaknesses.

---

> ### Author Response · Authors · 2024-11-23
> **Response to Reviewer msia I**
>
> We thank the reviewer for recognizing the usefulness of our derived PAC-Bayesian bounds on our theoretical results. We believe that there are misunderstandings about the soundness of our results. If you agree that we've managed to address all issues, please consider raising the score. But if there are any remaining concerns, please do not hesitate to follow up - we would be more than happy to address your further questions.
>
> > The authors consider the Hessian of the loss function $\ell(\theta , x+\delta ,y)$ in Lines 204-205.
>
> We've clarified the notation $\delta$ in our General Response 1 and demonstrated why it will not affect the validity of our theoretical results in our General Response 2.
>
>
> > Can the authors clarify how they compute the Hessian of the adversarial loss function $\ell(\theta , x+\delta ,y)$?
>
> > In the definition of WCI in Equation (5), I wonder how the authors compute $\text{Tr}(H_k)$.
>
> Below, we answer the reviewer's questions regarding the computation of the Hessian matrix $\mathbf{H}$. Let $\mathbf{H}\_k$ be the Hessian matrix for the $k$-th layer of the neural network $h\_\theta$. A short answer is that we empirically compute the Hessian matrix and its trace by employing projected gradient descent (PGD) to find an approximated perturbation to the worst-case (i.e., $\delta_{\mathrm{pgd}}$) and fix the perturbation as $\delta$ in the loss function when computing the Hessian.
>
>
> Theoretically, $\mathbf{H}\_k$ (see Equation 10 in the appendix) is defined as:
> $$
> \mathbf{H}\_k = \nabla^2\_{\theta} \ \mathbb{E}\_{(x, y) \in \mathcal{S}}\big[ \ell(\theta, x + \delta(\theta,x,y), y) \big ] = \sum\_{i,j} \frac{\partial^2 \mathcal{L}\_{\mathcal{S}}(\theta, x + \delta(\theta,x,y), y)}{\partial W\_k[i,j]^2},
> $$
> where $\delta(\theta, x, y)$ denotes the worst-case perturbation (its dependence on $\theta$ and $(x,y)$ is clarified in our General Response 1) and $W_k$ denotes the weights of the $k$-th layer of the neural network. For simplicity, we write $\delta(\theta) = \delta(\theta,x,y)$ in the following discussions. The trace of the Hessian matrices are introduced in Lemma 2.4 to simplify the _Classifier Variability_ term in Equation 1 of our work.
>
> In our numerical analysis and experiments, however, the Hessian computation deviates from the idealized theoretical formulation, due to the challenges of optimizing over both $\delta$ and $\theta$ simultaneously. Thus, we resort to approximation and optimization tools, such as PGD to approximate the worst-case perturbation $\delta$, to compute the Hessian matrix and its trace. Specifically, our implementation proceeds in two distinct stages: (a) use Projected Gradient Descent (PGD) to obtain $\delta_{\mathrm{pgd}}$ that serves as a good approximation of the worst-case perturbation $\delta$ with respect to the corresponding $\ell_p$-norm ball, and (b) use auto differentiation to compute the Hessian by fixing the input perturbation as $\delta_{\mathrm{pgd}}(\theta)$ (i.e., the Hessian is computed with respect to $\mathcal{L}\_{\mathcal{S}}(\theta, x + \delta\_{\mathrm{pgd}}(\theta), y)$.
>
> There may exist a gap between the theoretic Hessian definition (Equation 10) and the empirically computed Hessian matrix used in our experiments. However, we believe the empirical Hessian matrix can largely preserve the Hessian structure with respect to the (worst-case) robust loss $\mathcal{L}\_{\mathcal{S}}(\theta, x + \delta(\theta), y)$ and will not affect the conclusion of our numerical analysis and experiments. In fact, PGD-based adversarial training is still a widely adopted method for adversarial robustness, thus using $\delta_{\mathrm{pgd}}$ is a reasonable choice for understanding robust generalization of adversarially-trained neural networks.

---

> ### Author Response · Authors · 2024-11-23
> **Response to Reviewer msia II**
>
> > Lemma 2.4 is based on an unspecified assumption that "the empirical loss $L_S(\theta ,x+\delta ,y)$ can be approximated using second-order Taylor expansion". Can the authors clarify the precise mathematical statement of the assumption?
>
> Below, we lay out the precise assumption of the second-order loss approximation that we used in Lemma 2.4. Let $\mathcal{L}\_{\mathcal{S}}(\theta, x+\delta(\theta), y)$ be the robust loss with $\delta$ being the worst-case perturabtion (clarified in our General Response 2). For any $\theta'$ sampled from the posterior distribution $\mathcal{Q}$, we assume that the empirical robust loss at $\theta'$ can be approximated by:
> $$\mathcal{L}\_\mathcal{S}(\theta', x + \delta(\theta'), y) \approx \mathcal{L}\_\mathcal{S}(\theta, x + \delta(\theta), y) + \tfrac{1}{2} \Delta \theta^\top \nabla^2\_{\theta} \mathcal{L}\_\mathcal{S}(\theta, x + \delta(\theta), y) \Delta \theta.$$
> Note that we keep the Taylor expansion terms up to the second-order terms. Here, the first-order term can be discarded because $\theta$ is considered to be at or near a stationary point of the robust loss function, which is a common setting considered in prior literature on deep learning generalization (see [1-2], for example).
>
> [1] Stephan, Mandt, Matthew D. Hoffman, and David M. Blei. ["Stochastic gradient descent as approximate bayesian inference."](https://arxiv.org/pdf/1704.04289) Journal of Machine Learning Research 18.134 (2017): 1-35.
>
> [2] Keskar, Nitish Shirish, et al. ["On large-batch training for deep learning: Generalization gap and sharp minima."](https://openreview.net/pdf?id=H1oyRlYgg) arXiv preprint arXiv:1609.04836 (2016).
>
> > First, standard ReLU activation functions are not twice-differentiable and so the Hessian analysis does not apply to them. Also, $L_S$ is supposed to be the adversarial loss, which means the authors assume that the worst-case loss $\max_{\Vert\delta\Vert\le \epsilon } \ell(\theta, x+\delta , y)$ has a bounded Hessian. Therefore, the flatness of the adversarial loss seems a very strong assumption which very likely would not work even using twice-differentiable activation functions.
>
> We would like to clarify that our analysis does **not** rely on the assumption that the adversarial loss $\ell(\theta, x+\delta(\theta), y)$ has a bounded Hessian or that the flatness of the adversarial loss holds universally.
>
> While ReLU activations are not twice-differentiable (due to the kink at zero), this does not invalidate the Hessian analysis. The Hessian is computed over the weights (parameters) of differentiable layers (e.g., linear or convolutional layers). The non-differentiability of ReLU affects the loss landscape indirectly but does not preclude meaningful second-order approximations in regions where the network behaves smoothly. In practice, deep networks often operate in regions where the activation paths induced by ReLU are stable. In these regions, the overall loss landscape can still be locally approximated using second-order methods.
>
> The second-order Taylor expansion assumption for the adversarial loss is a simplification that may not strictly hold in all cases, particularly for networks with ReLU activations or sharp adversarial loss surfaces. Despite the non-differentiability of ReLU, second-order approximations have been shown empirically to align with network behaviors in both adversarial and standard training scenarios. This includes works that analyze sharpness, curvature, and generalization bounds (see Equation 8 in [3]). In practice, it has been shown to capture meaningful characteristics of the loss landscape in regions where the network behaves smoothly.
>
>
> [3] Wu, Dongxian, Shu-Tao Xia, and Yisen Wang. ["Adversarial weight perturbation helps robust generalization."](https://proceedings.neurips.cc/paper_files/paper/2020/file/1ef91c212e30e14bf125e9374262401f-Paper.pdf) Advances in neural information processing systems 33 (2020): 2958-2969.

---

> ### Author Response · Authors · 2024-11-23
> **Response to Reviewer msia III**
>
> > Also, when I checked the proof of this lemma, I could not understand Lines 804-807. How does the "second-order Taylor series expansion around $\theta$" yield Equation (8)?
>
> Equation 8 directly follows by expanding the loss difference using the second-order Taylor expansion at $\theta$ and leveraging the properties of Gaussian-distributed perturbations to bound the expected loss difference. Below, we explain the detailed derivations steps:
>
> - Step 1. Because of the second-order loss approximation, we obtain
> $$\mathcal{L}\_\mathcal{S}(\theta', x + \delta(\theta'), y)] - \mathcal{L}\_{\mathcal{S}}(\theta, x + \delta(\theta), y)\approx \frac{1}{2}\Delta\theta^\top \nabla^2\_{\theta} \big[ \mathcal{L}\_{\mathcal{S}}(\theta, x + \delta(\theta), y) \big ] \Delta\theta,$$
> where $\Delta = \theta' - \theta$ follows $\mathcal{N}(0,\Sigma)$ because we set the posterior $\mathcal{Q}$ as $\mathcal{Q} = \mathcal{N}(\theta, \Sigma)$.
>
> - Step 2. By taking expectation over $\theta'\sim\mathcal{Q}$ with respect to the above, we derive
> $$\mathbb{E}\_{\theta'\sim\mathcal{Q}}\mathcal{L}\_\mathcal{S}(\theta', x + \delta(\theta'), y)] - \mathcal{L}\_{\mathcal{S}}(\theta, x + \delta(\theta), y)\approx \frac{1}{2}\mathbb{E}\_{\Delta\theta\sim\mathcal{N}(0,\Sigma)}\Delta\theta^\top \nabla^2\_{\theta} \big[ \mathcal{L}\_{\mathcal{S}}(\theta, x + \delta(\theta), y) \big ] \Delta\theta.$$
>
> We will revise the proof of Lemma 2.4 in the revised manuscript to ensure clarity and rigor to eliminate any typos, ambiguities or confusion. Specifically, we will carefully review all mathematical expressions, assumptions, and steps to ensure they are accurate and align with the intended derivation shown above. Your feedback is invaluable, and we are committed to providing a clear and precise explanation of Lemma 2.4 in the final version.

---

> ### Author Response · Authors · 2024-11-29
> **Additional Response to Reviewer msia**
>
> Thank you once again for your constructive comments. We have implemented several key modifications in the manuscript to address your feedback:
>
> Definition 2.1 (Adversarial Risk and $\delta$): Modified in Section 2, Lines 119-124, to clarify the dynamic optimization of $\delta$ for each input example.
>
> Lemma 2.4 (Hessian Calculation): Detailed computational methods now included in Appendix A.3, Lines 932-937, emphasizing our approach to second-order loss approximation.
>
> Empirical Loss Approximation: Expanded discussion in Section 2.2, Lines 220-225, about the conditions and justification for using the Taylor expansion.
>
> We believe these revisions address the concerns you raised and enhance the manuscript’s technical depth. We look forward to your thoughts on these updates and any further suggestions you might have.

---

> > ### Comment · Reviewer_msia · 2024-11-30
> >
> > I thank the authors for their responses to my questions and for providing clarifications. As I explain in the following, I believe the authors' approach to computing Hessian is wrong, and the computed Hessian in the experiments can be significantly different from the Hessian of the target function in adversarial training. I therefore keep my score as I think this is a fatal error. I am open to discuss with the authors if they think I miss anything here.
> >
> > > "A short answer is that we empirically compute the Hessian matrix and its trace by employing projected gradient descent (PGD) to find an approximated perturbation to the worst-case (i.e.,
> > ) and fix the perturbation as in the loss function when computing the Hessian."
> >
> > I believe this approach leads to an incorrect Hessian when applied to a function of the form $$g(\theta) = \max_{\delta} L(\delta , \theta)$$
> > Note that this is not because the authors use PGD for solving the maximization. The reason is that unlike first-order derivative (gradient), the second-order derivative (Hessian) of the max function $g(\theta)= \max_{\delta} L(\delta , \theta)$ is not equal to the Hessian of the objective function $L(\delta^* , \theta)$ at the optimal perturbation $\delta = \delta^*(\theta)$. As said, this result is only true for the first-order derivative, as it is correctly used in adversarial training (Check Danskin's theorem in (Madry et al, 2017)).
> >
> > To show why this is not the case, let me give a simple counterexample $$L(\delta , \theta) = \delta^\top \theta - \frac{1}{2}\Vert \delta\Vert^2_2$$ For a fixed $\delta$, this function $L(\delta , \theta)$ is linear in $\theta$ and so its Hessian is zero: $\nabla^2_{\theta} L(\delta , \theta) = \mathbf{0}$. Therefore, if you solve the maximization $\max_{\delta} L(\delta , \theta)$ to find $\delta^* = \delta^*(\theta)$, and then you compute the Hessian of  $L(\delta^* , \theta)$ while fixing $\delta^*$, the Hessian will be zero.
> >
> > However, the max function has indeed a non-zero Hessian. You can easily see that
> > $$\frac{1}{2}\Vert\theta\Vert^2_2 = \max_{\delta}\Bigl[ L(\delta , \theta) = \delta^\top \theta - \frac{1}{2}\Vert \delta\Vert^2_2\Bigr]$$
> > Therefore, the correct hessian of $g(\theta) = \max_{\delta} L(\delta , \theta)$ is the identity matrix $\nabla^2_{\theta} g(\theta) = I_{d}$. This counterexample shows why the Hessian of the max function cannot be computed similar to the gradient, a mistake that authors seem to have made in the Hessian calculation.

---

> > > ### Author Response · Authors · 2024-12-01
> > > **Additional Response to Reviewer msia**
> > >
> > > We appreciate the reviewer’s concerns. To clarify, in our approach, we **do not fix** $\delta^*$ **as a value but as a function of $\theta$** in our numerical implementation. Using `torch.autograd.grad` with `create_graph=True`, we ensure that the **dependency of** $\delta^*$ **on** $\theta$ **is preserved during the Hessian computation**. Specifically, the Hessian is computed **before** `robust_loss.backward()` to ensure all second-order terms are correctly captured.
> > >
> > > Additionally, we validated our method using the reviewer’s counterexample $L(\delta, \theta) = \delta^\top \theta - 0.5 \|\delta\|_2^2$. By applying the same numerical approach, specifically `torch.autograd.functional.hessian(delta_max_function, theta)`, we correctly obtained the Hessian as the identity matrix $\mathbf{I}_d$, **not a zero matrix**. This confirms the correctness of our implementation and refutes the claim of fixing $\delta^*$ improperly.

---

> > > > ### Comment · Reviewer_msia · 2024-12-01
> > > >
> > > > I read the author's response, which is opposite to what they had originally stated in response to my earlier question on computing Hessian. I am quoting their original response here for reference:
> > > >
> > > > > "A short answer is that we empirically compute the Hessian matrix and its trace by employing projected gradient descent (PGD) to find an approximated perturbation to the worst-case (i.e., ) and **fix the perturbation as in the loss function** when computing the Hessian."
> > > >
> > > > Also, can I ask the authors to at least share the code block for computing Hessian for the case with PGD attack? The code is not provided in the submission and it is not possible to validate the claim.

---

> > > > > ### Comment · Reviewer_YVUF · 2024-12-01
> > > > > **Careful about using only single permutation**
> > > > >
> > > > > Dear authors, chiming into the discussion, reviewer msia is correct to be concerned about the hessian computation because there is generally not a unique maximizer  of the adversarial problem in neural nets, as demonstrated (and fixed, if inefficiently) in https://openreview.net/forum?id=I3HCE7Ro78H . The TL;DR; is that a single adv-exp yields a subgradient, not a gradient, and one has to choose a safe direction from the subdifferential (in the reference, this is done by choosing the minimum norm convex combination). This ensures safe descent.
> > > > >
> > > > > Extending this to the hessian could be challenging, although possible. Maybe your analysis can consider the bordered hessian instead?

---

> ### Author Response · Authors · 2024-12-01
> **Additional Response to Reviewer msia I**
>
> We appreicate the reviewer's follow up comments and apologize for any confusion caused by our previous clarification. In the following, we clarify why the theoretical Hessian matrix can be estimated by first finding an approximated solution $\delta\_{\mathrm{pgd}}$ to the inner maximization problem using PGD then computing the second-order derivative with respect to the loss function at $(x+\delta\_{\mathrm{pgd}}, y)$. In particular, we will explain both from the general perspective based on the Danskin's Theorem and using the specific example provided by the reviewer.
>
> Consider $\ell\_\infty$ perturbations bounded by $\epsilon$ and the corresponding inner maximization problem with respect to $(x,y)$. According to the Danskin's Theorem, we can obtain:
> $$
>     \nabla\_{\theta} \bigg[ \max\_{||\delta||\_\infty \leq \epsilon} \ell(\theta, x+\delta, y) \bigg] = \nabla\_{\theta} \big[ \ell(\theta, x+\delta^*, y) \big]\approx \nabla\_{\theta} \big[ \ell(\theta, x+\delta\_{\mathrm{pgd}}, y) \big],
> $$
> where $\delta^* = {\mathrm{argmax}}\_{||\delta||\_\infty\leq \epsilon} \: \ell(\theta, x+\delta, y)$ denotes the worst-case perturbation and $\delta\_{\mathrm{pgd}}$ denotes the perturbation returned by PGD attack. As the reviewer mentioned, the above equation characterizes the theoretical basis of adversarial training. Taking the derivative with respect to $\theta$ for both sides of the above equation, we immediately have:
> $$
>     \nabla\_{\theta}^2 \bigg[ \max\_{\|\delta\|\_\infty \leq \epsilon} \ell(\theta, x+\delta, y) \bigg] = \nabla\_{\theta}^2 \big[ \ell(\theta, x+\delta^*, y) \big] \approx \nabla\_{\theta}^2 \big[ \ell(\theta, x+\delta\_{\mathrm{pgd}}, y) \big].
> $$
> This suggests that to estimate the theoretical Hessian matrix with the worst-case perturbation $\delta^*$, one can first find an approximated solution $\delta_{\mathrm{pgd}}$ using PGD attack then compute the Hessian matrix with respect to the loss function at $(x+\delta_{\mathrm{pgd}}, y)$. We want to emphasize that both $\delta^*$ and $\delta_{\mathrm{pgd}}$ should **not be understood as fixed constant vectors**, since their values essentially depend on the model parameter $\theta$ as the input.

---

> ### Author Response · Authors · 2024-12-01
> **Additional Response to Reviewer msia II**
>
> Below, we further illustrate why this is the case using the example provided by the reviewer. Consider the following optimization  problem:
> $$
> \max\_{\delta} L(\theta, \delta), \quad \text{where} \quad L(\theta, \delta) = \delta^{\top}\theta - \frac{1}{2}||\delta||\_2^2.
> $$
> Theoretically, we know the optimal solution is $\delta^* = \theta$ and we immediately have:
> $$
> \nabla^2\_\theta \bigg[\max\_\delta L(\theta, \delta)\bigg] = \nabla^2\_\theta \ L(\theta, \delta^*) = \nabla\_\theta^2 \left( \frac{1}{2} ||\theta||\_2^2 \right) = \mathbf{I},
> $$
> suggesting the theoretical Hessian is an identity matrix. To simulate PGD attack used for approximating the worst-case, we implement a simple gradient ascent algorithm to solve the maximization problem $\max_\delta L(\theta, \delta)$, then compute the Hessian matrix with respect to the $L(\theta, \hat\delta)$ with $\hat\delta$ denoting the gradient descent returned solution. We verify that $\nabla^2_{\theta} L(\theta, \hat\delta)$ is very close to an identity matrix (instead of a zero matrix), which confirms the validity of our numerical computation of the Hessian matrix. We provide Python code of this implementation for the reviewer to check:
> ```
> import torch
>
> def counterexample_test():
>     # Define the counterexample function L(delta, theta)
>     # L(delta, theta) = delta^T * theta - 0.5 * ||delta||_2^2
>     def loss_function(delta, theta):
>         return torch.dot(delta, theta) - 0.5 * torch.norm(delta, p=2) ** 2
>
>     # Compute the optimal perturbation delta^* as a function of theta
>     def compute_delta_star(theta, max_iters=10, lr=0.9):
>         delta = torch.zeros_like(theta, requires_grad=True)  # Initialize delta
>         for _ in range(max_iters):
>             loss = loss_function(delta, theta)  # Calculate the current loss
>             grad = torch.autograd.grad(loss, delta, create_graph=True)[0]  # Calculate the gradient of loss with respect to delta
>             delta = delta + lr * grad  # Gradient ascent update delta
>         return delta
>
>     # Calculate g(theta) = max_delta L(delta, theta)
>     def max_function(theta):
>         delta_star = compute_delta_star(theta)
>         return loss_function(delta_star, theta)
>
>     # Initialize theta
>     theta = torch.rand(5, requires_grad=True)
>
>     # Compute the Hessian using numerical methods
>     hessian = torch.autograd.functional.hessian(max_function, theta)
>
>     print(hessian)
>
> if __name__ == "__main__":
>     counterexample_test()
> ```
> Even if we lower the value of `max_iters` or `lr`, the result will never be $\mathbf{0}$. It only affects the size of the error between the numerical calculation and the actual value.
>
> We believe the above example and explanations should clarify the reviewer's concern about the validity of our Hessian computation regarding the adversarial loss. For the sake of completeness, we explain how the Hessian matrix is computed in our experiments. In particular, we calculate the hessian following the lines 311 to 333 in the [code repository of the "Robust Overfitting" paper (Rice et al., 2020)]( https://github.com/locuslab/robust_overfitting/blob/master/train_cifar.py).
> ```
> robust_output = model(normalize(torch.clamp(X + delta[:X.size(0)], min=lower_limit, max=upper_limit)))
> robust_loss = criterion(robust_output, y)
> ```
> Then in line 334 we add a function to compute WCI as well as hessian:
> ```
> WCI, WCI_norm, WCI_trace = funcWCI(model, robust_loss)
> ```
> The specific content of this function is as follows:
> ```
> def funcWCI(model, loss, epsilon=1e-6):
>     sum_WCI = 0
>     sum_norm = 0
>     sum_trace = 0
>
>     grads = torch.autograd.grad(loss, [p for p in model.parameters() if p.requires_grad], create_graph=True)
>
>     for param, grad in zip(model.parameters(), grads):
>         if grad is not None and torch.isfinite(grad).all() and grad.abs().sum() > epsilon:
>             grad_clipped = torch.clamp(grad, -1e6, 1e6)
>             grad2 = grad_clipped.pow(2) + epsilon
>             trace = torch.autograd.grad(grad2.sum(), param, retain_graph=True)[0]
>             if trace is not None and torch.isfinite(trace).all():
>                 hessian_trace = trace.sum()
>                 layer_frobenius = torch.norm(param, p='fro') ** 2
>                 WCI = (layer_frobenius * hessian_trace + epsilon).sqrt()
>
>                 # Collecting layer-specific details
>                 if param.requires_grad:
>                     if torch.isfinite(WCI):
>                         sum_WCI += WCI
>                     sum_norm += layer_frobenius
>                     sum_trace += hessian_trace
>
>     return sum_WCI, sum_norm, sum_trace
> ```

---

> ### Author Response · Authors · 2024-12-01
> **Response to Reviewer YVUF**
>
> Thanks for pointing out the additional reference and the challenges for Hessian computation, particularly in cases where the adversarial maximizer is not unique. We recognize the validity of the concern that a single adversarial example can yield a subgradient rather than a true gradient, as highlighted in the referenced work. The proposed approach of selecting a minimum norm convex combination from the subdifferential indeed ensures safe descent and could be a robust strategy for handling subgradient challenges. Extending such approaches to the Hessian level, including the use of a bordered Hessian, is an interesting direction for future work.
>
> Nevertheless, we want to emphasize that the numerical Hessian matrix computation used in our paper is valid, as we clarified in our new response to Reviewer msia. In paricular, we explain why such a computation is reasonable assuming the Danskin's Theorem is applicable (i.e., the optimal solution to the inner-maximization problem is unique) and validate numerically using a gradient ascent approximation with respect to a specific example $\max_{\delta} \left(\delta^{\top}\theta - \frac{1}{2}||\delta||_2^2\right)$. As the authors of https://openreview.net/forum?id=I3HCE7Ro78H said, PGD and DDi seem to behave similarly in the later stages of training. So here, we use PGD as the basis, given its wide adoption in adversarial training and its practical efficiency. Our approach aligns with standard practices and has been shown to work robustly in the settings we analyze.
>
> While we appreciate the suggestion to consider bordered Hessians, which could provide additional theoretical guarantees in addressing subgradient issues, we believe our current method is accurate and efficient for the scope of our study. Future work could further explore these ideas to generalize the analysis to settings where non-uniqueness is a significant concern. Thanks again for raising this important point and for the helpful reference. It has been a valuable addition to this discussion.

---

### Author Response · Authors · 2024-11-23
**General Response I**

We thank all the reviewers for their detailed feedback and insightful comments. In particular, we are delighted that all four reviewers acknowledged key aspects of our work: Reviewer YVUF praised the **novelty** of combining PAC-Bayesian analysis with Sharpness-Aware Minimization (SAM) and highlighted the clarity and structure of our presentation; Reviewer tWVD appreciated the **quality** of our theoretical derivations and the detailed documentation of our methodology; Reviewer msia recognized the **potential value** of employing PAC-Bayesian generalization bounds in the context of adversarial training; Reviewer 5E9W commended the detailed description of our experimental settings and acknowledged the promise of our learning rate scheduler in mitigating robust overfitting.

Below, we provide general responses to the concerns of the reviewers regarding the notations and the soundness of our theoretical results, which we believe are most important to clarify. Other questions and comments will be clarified in the corresponding individual responses. We anticipate an interactive conversation, so we will be more than happy to address any of your remaining questions or conerns. An updated version of our paper will be released later to include our discussions during the rebuttal period.

> 1. Questions about Definition 2.1, especially the notation $\delta$

We admit that the notation $\delta$ in Definition 2.1 could have been clearer, especially about its dependence on $\theta$ and $(x,y)$. To clarify, we lay out the definition of $\delta$ in the most explicit form. Let $\mathcal{B}\_\epsilon(0)=\{\delta^{\prime}\in\mathcal{X}:\Delta(\delta^{\prime},0)\leq\epsilon\}$ be the perturbation ball centered at $0$ with distance metric $\Delta:\mathcal{X}\times\mathcal{X}\to\mathbb{R}\_{\geq0}$ and strength $\epsilon>0$. Given a classifier $h_\theta$, for any input example $(x,y)$, $\delta$ is defined as the worst-case perturbation within the $\epsilon$-ball $\mathcal{B}\_\epsilon(0)$ such that the loss function with respect to $h_\theta$ at $(x,y)$ is maximized:
$$
\delta = \delta(\theta, x, y) = \underset{\delta' \in \mathcal{B}_{\epsilon}(0)}{\arg\max} \ \ell(\theta, x + \delta, y),
$$
where $\ell$ denotes the individual loss function such as cross-entropy loss. We will clarify this notation in the revised version of our paper to avoid potential confusions.

> 2. The notation $\delta$ may have caused errors in the theoretical analysis. The main concern is that the theoretical results are not convincing, or underdescribed at least.

We understand the unclear notation $\delta$ might lead to this concern, but we believe this is a misunderstanding that we hope to clarify. Below, we explain why the notation $\delta$ (being dependent on $\theta$ and $(x,y)$) does not affect the conclusion of our theoretical results. The main reason is that our theoretical analysis and the derived adversarially robust generalization bound (Equation 1) is based on the PAC-Bayesian framework, which is generic in terms of the prior distribution $\mathcal{P}$, posterior distribution $\mathcal{Q}$ and the loss function that the model aims to minimize. That being said, explicitly writing $\delta = \delta(\theta, x, y)$ does not affect the soundness of any of our theoretical results, including Lemma 2.2, Lemma 2.3, Lemma 2.4 and Theorem 2.5. To explain this more clearly, we provide the following justifications for the reviewers to better understand the essence of our theoretical results.

The standard PAC-Bayesian generalization bound (see Theorem 2.1 in [1]) states that for any posterior distribution $Q$, the following inequality holds with high probability:
$$
    \mathbb{E}\_{\theta\sim\mathcal{Q}} \mathbb{E}\_{(x,y)\sim \mathcal{D}}[\ell(\theta, x , y)] \leq  \mathbb{E}\_{\theta\sim\mathcal{Q}} [\mathcal{L}\_{\mathcal{S}}(\theta, x , y)] + \frac{1}{\lambda} \mathrm{KL}(\mathcal{Q} || \mathcal{P}) + \text{const.},
$$
where $\mathbb{E}\_{\theta\sim\mathcal{Q}} \mathbb{E}\_{(x,y)\sim \mathcal{D}}[\ell(\theta, x , y)]$ represents the expected population loss of classifiers $h_\theta$ with $\theta$ sampled from the posterior distribution $Q$, $\mathbb{E}\_{\theta\sim\mathcal{Q}} [\mathcal{L}\_{\mathcal{S}}(\theta, x , y)]$ captures how well $h\_\theta$ (with $\theta\sim\mathcal{Q}$) fits the finite-sample set $\mathcal{S}$, and $\mathrm{KL}(\mathcal{Q} || \mathcal{P})$ is the KL divergence between posterior and prior which regularizes the overfitting.

---

> ### Author Response · Authors · 2024-11-23
> **General Response II**
>
> An important observation is that for the above PAC-Bayesian generalization bound to hold, there are __no__ assumption imposed on the loss function $\ell$, thus it directly applies to the adversarial loss (with respect to the worst-case perturbation $\delta$). In fact, the only required assumption is that the prior distribution $\mathcal{P}$ is data independent such that the Fubini's theorem can be applied to exchange the intergration with $\theta$ and the sample (see Equation 2.2 in [1]). Therefore, when deriving robust generalization bound, we can simply replace the standard loss in the above inequality by their robust counterpart. More specificaly, denote by $\ell\_{\mathrm{rob}}(\theta, x, y) = \underset{\delta' \in \mathcal{B}\_{\epsilon}(0)}{\max} \ell(\theta, x+\delta', y)$ the individual robust loss with the worst-case perturabtion, then we have
> $$ \mathbb{E}\_{\theta\sim\mathcal{Q}} \mathbb{E}\_{(x,y)\sim \mathcal{D}}[\ell_{\mathrm{rob}}(\theta, x , y)] \leq  \mathbb{E}\_{\theta\sim\mathcal{Q}} \bigg[\frac{1}{|\mathcal{S}|}\sum\_{(x,y)\in\mathcal{S}}\ell\_{\mathrm{rob}}(\theta, x , y)\bigg] + \frac{1}{\lambda} \mathrm{KL}(\mathcal{Q} || \mathcal{P}) + \text{const.}, $$
> This lays the foundation for Lemma 2.2 in our paper, which establishes a generic robust generalization bound that holds for any classifer $h_\theta$ (not necessary neural network), posterior distribution $\mathcal{Q}$ and data-independent prior $\mathcal{P}$.
>
> To extract meaningful insights from the generic robust generalization bound, however, we need to set $\mathcal{P}$ and $\mathcal{Q}$ properly such that the generalization bound is tight and can be simplified into analytical terms. These are exactly what we have done in Lemma 2.3, Lemma 2.4 and Theorem 2.5, where we set $\mathcal{P}$ as $\mathcal{N}(0, \sigma^2 \Sigma)$ and $\mathcal{Q}$ as $\mathcal{N}(\theta, \Sigma)$ with covariance matrix $\Sigma = \text{diag}(\sigma_1^2 \mathbf{I}_1, \sigma_2^2 \mathbf{I}_2, \dots, \sigma_K^2 \mathbf{I}_K)$ to simplify the terms (Lemma 2.3 and Lemma 2.4), and then choose the variance for the $k$-th layer $\sigma_k^2 = \sqrt{\|\mathbf{W}_k\|_F^2 / (2\lambda \mathrm{Tr}(\mathbf{H}_k))}$ to minimize the bound (Theorem 2.5). Selecting Gaussian prior and posterior aligns well with existing literature that employs the PAC-Bayesian framework (see [2,3]). Such choices of $\mathcal{P}$ (which is clearly data-dependent) and $\mathcal{Q}$ do not violate the soundness of our theoretical results, again because of the generic nature of PAC-Bayesian framework.
>
> [1] Pierre Alquier et al., ["User-friendly introduction to pac-bayes bounds"](https://arxiv.org/pdf/2110.11216). Foundations and Trends in Machine Learning, 17(2):174–303, 2024.
>
> [2] Mbacke, Sokhna Diarra, Florence Clerc, and Pascal Germain. ["PAC-Bayesian generalization bounds for adversarial generative models."](https://proceedings.mlr.press/v202/mbacke23a/mbacke23a.pdf) International Conference on Machine Learning. PMLR, 2023.
>
> [3] Jin, Gaojie, et al. ["Enhancing adversarial training with second-order statistics of weights."](https://openaccess.thecvf.com/content/CVPR2022/papers/Jin_Enhancing_Adversarial_Training_With_Second-Order_Statistics_of_Weights_CVPR_2022_paper.pdf) Proceedings of the IEEE/CVF conference on computer vision and pattern recognition. 2022.

---

> > ### Author Response · Authors · 2024-11-23
> > **General Response III**
> >
> > > 3. The potential gap from the "ideal" robust generalization bound
> >
> > We hope the above explanations have clarified why our PAC-Bayesian robust generalization bounds are sound. To complete the picture, we explain the "ideal" robust generalization bound one might want to prove and shed light on the potental future directions of our work. Ideally, we want to prove the _tightest possible_ robust generalization bound with respect to _some specific family of ML models_, e.g., neural networks with certain architecture that are trained using stochastic gradient descent. Specifically for understanding the robust overfitting phenomenon, proving a generic PAC-Bayesian robust generalization bound is definitely useful for characterizing the key model indicator that generally connects with adversarially robust generalzation. However, one might argue that a tighter robust generalization bound can be derived, if we make use of the specific information of model architecture and even the underlying characteristics of adversarial training algorithms, potentially suggesting more insights on what factors cause robust overfitting of adversarially-trained models and how to improve model robustness. This could be an interesting future direction of our work. Nevertheless, to the best of our knowledge, this remains an open problem to the field, even for standard deep learning generalization. Addressing this problem requires advancements not only in theoretical frameworks but also in a deeper understanding of the interactions between model architectures, training algorithms, and adversarial robustness. Therefore, we believe that our theoretical results, the discovery of WCI and its strong ability of capturing the model's robust generalization are important contributions for advancing the field.

---

### Author Response · Authors · 2024-11-29
**Response to Reviewers' Comments**

We sincerely appreciate the valuable feedback provided by the reviewers. In response to the comments, we have made __several substantial enhancements__ to our manuscript to clarify our methodology, strengthen the theoretical foundation, and validate our claims with additional empirical evidence. Below, we summarize the key changes made to the manuscript:

__1. Enhancement of the Related Work Section:__ We have expanded the discussions of relevant literature in the Related Work section. This includes a more comprehensive review of existing methods and findings pertinent to adversarially robust generalization, which now better contextualizes our contributions within the field.

__2. Clarifications in Theoretical Descriptions:__ We have clarified the definitions and implications of several key theoretical concepts, including the delta in adversarial perturbations, the PAC-Bayesian framework, and Second-Order Loss Approximation. These clarifications aim to make the theoretical aspects of our work more accessible and understandable to readers.

__3. Additional Experimental Results:__ In response to the reviewers' request for empirical validation, we have conducted additional experiments demonstrating a strong positive correlation between our proposed Weight-Curvature Index (WCI) and the generalization gap in adversarially trained models. These results further substantiate the practical relevance and effectiveness of the WCI as a robust generalization measure.

__4. Inclusion of AutoAttack Evaluation:__ To address concerns regarding the robustness and reliability of our findings, we have included results from evaluations using the AutoAttack framework, which is known for its rigorous assessment of model robustness. This addition provides a more robust validation of our model's performance against sophisticated adversarial attacks.

These modifications have significantly strengthened our manuscript by providing clearer theoretical insights, more comprehensive literature integration, and robust empirical support for our claims. We believe that these changes address the reviewers' concerns effectively and enhance the manuscript's contribution to the field of adversarial machine learning.

We are grateful for the opportunity to improve our work based on the reviewers' insightful comments.

---

### Meta-Review · Area_Chair_8nXk · 2024-12-16

**Metareview:**

This paper introduces the Weight Curvature Index (WCI) as a metric to evaluate the robust generalization error via norm-bounded adversarial perturbations under the PAC-Bayesian framework. The derived generalization bound is based on a second-order loss approximation and realted to the WCI score.

The main issue of this submission is how to correctly compute the Hessian of the function $g(\theta) = \max_{\delta} f(\theta, \delta)$ when the optimization problem is non-convex. It requires an accurate approximation of the Jacobian of $\delta^*(\theta)$ (optimal solution given $\theta$), for which even computing the value is challenging due to the non-convexity of the neural network optimization.

This submission received extensive discussions between reviewers and AC. We believe that this issue might be considered carefully and suggest the author to fix this issue for the next-round submission.

**Additional Comments On Reviewer Discussion:**

We discuss about the how to correctly compute the Hessian of the function $g(\theta) = \max_{\delta} f(\theta, \delta)$ when the optimization problem is non-convex. This is not a trivial issue in this submission and we believe that this issue should be addressed rigiously and formally.

---

### Decision · Program_Chairs · 2025-01-22

Reject